# Interplay between T3SS effectors, ExoY activation, and cGMP signaling in *Pseudomonas aeruginosa* infection

Vincent Deruelle [1] ✉, Gabrielle Dupuis[1,2], Dorothée Raoux-Barbot[1], Elysa Lim[1,4], Roberto Ponce-López[1], Magda Teixeira Nunes [3], Daniel Ladant [1], Louis Renault [3] & Undine Mechold[1]

The Type III Secretion System (T3SS) of *Pseudomonas aeruginosa* injects effector proteins into host cells to subvert cellular processes and promote infection. While the roles of several T3SS effectors in virulence are well established, the function of the nucleotidyl cyclase ExoY has remained elusive and debated. Here, we show that ExoY-produced cyclic GMP (cGMP) regulates the cytotoxic activity of the co-injected effector ExoT, thereby modulating host cell damage. Using engineered *P. aeruginosa* strains expressing ExoY variants with distinct substrate preferences, we demonstrate that cGMP production by ExoY limits ExoT-mediated dephosphorylation of the host adaptor protein CrkII, a downstream consequence of ExoT's ADP-ribosyltransferase activity. This attenuation reduces ExoT-induced cell retraction and decreases bacterial virulence. In contrast, we find that ExoT and another T3SS effector, ExoS, can inhibit ExoY activity and cGMP production in certain cell types, thus limiting ExoY's regulatory influence. These findings highlight the intricate interplay between T3SS effectors within host cells and reveal an unrecognized layer of ExoY-based complexity in *P. aeruginosa*'s pathogenic strategy.

*Pseudomonas aeruginosa* is a highly adaptable Gram-negative opportunistic pathogen that infects a wide range of hosts. In humans, *P. aeruginosa* is a major cause of nosocomial infections, leading to acute or chronic life-threatening infections[1,2]. Lung infections caused by *P. aeruginosa* are particularly common in hospitalized patients, with clinical outcomes ranging from rapidly fatal pneumonia in immunocompromised patients to long-term bronchitis in cystic fibrosis (CF) patients[3–6]. The type 3 secretion system (T3SS) plays a key role in *P. aeruginosa* pathogenicity and its expression is associated with severe clinical outcomes and increased mortality[7–9]. This system delivers four effectors, ExoS, ExoT, ExoU, and ExoY, directly into the host cell's cytosol[10], where they are activated by specific host factors[11–14]. Most strains inject up to three exotoxins simultaneously. ExoS and ExoU were previously thought to be mutually exclusive[15,16], but recent findings have shown that some clinical isolates carry both genes and express them jointly, resulting in significantly enhanced virulence[17–19]. Initially identified as a T3SS effector with adenylate cyclase activity[20], ExoY was later characterized as a nucleotidyl cyclase (NC) with broad substrate specificity[21,22]. In host cells, ExoY is activated by F-actin[14,23–26] and promotes the accumulation of various cyclic nucleotide monophosphates (cNMPs), favoring cyclic guanosine monophosphate (cGMP) and cUMP rather than cAMP and cCMP[27]. In vitro, the substrate's preference of ExoY follows: GTP > ATP ≥ UTP > CTP[22], distinguishing it from ExoY-like orthologs and the structurally-related NC

[1]Institut Pasteur, Université Paris Cité, CNRS UMR3528, Biochemistry of Macromolecular Interactions Unit, Paris, France. [2]Centre de Recherche Saint-Antoine (CRSA), Sorbonne Université, INSERM UMR S 938, 5PMed: Pulmonary diseases, Pathogens, Physiopathology, Phenogenomics and Personalized Medecine, Paris, France. [3]Université Paris-Saclay, CEA, CNRS, Institute for Integrative Biology of the Cell (I2BC), Gif-sur-Yvette, France. [4]Present address: School of Biomolecular and Biomedical Science and Michael Smurfit Graduate Business School, University College Dublin, Dublin, Ireland. ✉e-mail: vincent.deruelle@pasteur.fr

toxins EF from *Bacillus anthracis* and CyaA from *Bordetella pertussis*, which prefer ATP[22,27,28].

The high prevalence of the *exoY* gene (89–100% of strains) and secretion of the protein across diverse isolates[15,17,29–31] suggest that ExoY is a key component of *P. aeruginosa* virulence. However, compared to ExoS, ExoT, and ExoU, which are known to directly induce cell death and facilitate infection, ExoY's role in *P. aeruginosa* pathogenicity remains debated. Discrepancies between studies[21,32–40] could result from the different experimental setups used to investigate ExoY, either using bacteria with a native expression level or using bacteria that overexpress ExoY from a multi-copy plasmid, which affects levels of ExoY injection[41] and can therefore conceal important functions or produce unphysiological effects.

The aim of this study is to investigate the contribution of ExoY-catalyzed cNMPs in the infection process of extracellular bacteria as *P. aeruginosa* remains primarily an extracellular pathogen, even though bacterial internalization has also been described[42,43]. We therefore constructed genetically modified *P. aeruginosa* strains expressing ExoY variants with different substrate specificities. This modification allowed us to identify a specific role for cGMP, and to uncover the mechanism underlying ExoY's antagonistic effect on ExoT cytotoxicity. Additionally, we found that ExoS or ExoT activities can counteract ExoY activation depending on the infected cell type, further emphasizing the intricate interplay among T3SS effectors in infected cells.

## Results

### Modification of the ExoY nucleotide-binding pocket to change substrate specificity

To investigate the role of ExoY-catalyzed cGMP production in infected cells, we engineered ExoY variants with altered nucleotide triphosphate (NTP) specificities, in order to shift substrate preference from GTP to ATP. Based on the recently solved structure of ExoY, we identified residues in the nucleotide-binding pocket potentially involved in GTP specificity[23] (Fig. 1a). These residues were replaced with those from structurally-related nucleotidyl cyclases (NCs) with narrower substrate specificities, such as the adenylate cyclase toxins from *Bordetella pertussis* (CyaA) and *Bacillus anthracis* (EF), or the ExoY-like modules from *Vibrio vulnificus* and *V. nigripulchritudo*[14,22,23] (Supplementary Fig. 1). Consequently, three ExoY variants were designed: ExoY[mut1] (N264V, F289V, S292G and D293A, one letter code for amino acids), ExoY[mut2] (F83L and S292G) and ExoY[mut3] (F83L, E258D and S292G). The recombinant proteins were expressed in *E. coli* and purified via an added C-terminal His-tag. The catalytic activities of each variant were assessed in vitro, using the purified proteins in the presence of the F-actin activator and radioactive GTP, ATP, UTP or CTP as substrate, and then compared to WT ExoY to illustrate the change in substrate specificity (Fig. 1b–e). A catalytically inactive mutant, ExoY[mut0] (K81M and K88I)[22] served as a negative control. As shown in Fig. 1, ExoY[mut1] exhibited a drastic reduction in cGMP and cUMP synthesis compared to WT ExoY but maintained cAMP and cCMP production. Conversely, ExoY[mut2] and ExoY[mut3], while retaining wild-type cUMP and cCMP production, produced significantly more cAMP than WT ExoY (38- and 23-fold increases, respectively). In addition, ExoY[mut3], which contains the additional E258D substitution compared to ExoY[mut2], showed a markedly reduced cGMP-synthetizing activity.

To confirm these biochemical data, we tested the substrate preferences of the mutants when expressed in NCI-H292 cells, a human bronchial epithelial cell line previously used to study *P. aeruginosa* infection[30]. For this, we cloned the WT and mutant *exoY* alleles into a Tet-On 3 G inducible expression plasmid, which enables a precise control of *exoY* transcription in cells by doxycycline. In parallel, we also created stable reporter cell lines that allow measurement of cGMP or cAMP accumulation by recording luminescence with the GloSensor technology (Promega) (Supplementary Fig. 2a). Plasmids (encoding the Tet-On 3 G transactivator and the various *exoY* alleles) were

therefore transfected into the cGMP/cAMP reporter cell lines to measure the production of cGMP or cAMP. Luminescence in doxycycline-induced cells was monitored and maximum values for each construct were plotted (Supplementary Fig. 2b–d). In the cGMP reporter cell line, only ExoY and ExoY[mut2] induced strong luminescence, while the luminescence in cells expressing the ExoY[mut1] and ExoY[mut3] was comparable to that of cells expressing the catalytically inactive ExoY mutant (ExoY[mut0]) (Fig. 1f). ExoY[mut2] and ExoY[mut3] produced significantly higher luminescence in the cAMP reporter cell line as compared to ExoY and ExoY[mut1] (Fig. 1g). Therefore, the cGMP/cAMP levels measured for the different ExoY alleles in transfected cells appeared to be in excellent agreement with the in vitro activities of the purified proteins.

Finally, we examined the cGMP/cAMP levels in cells infected with recombinant *P. aeruginosa* strains expressing the different ExoY variants. To maintain native expression level, we performed allelic exchange to replace the WT *exoY* allele and introduce mutations corresponding to each ExoY variant into the genome of *P. aeruginosa* PAO1FΔS(T,Y). NCI-H292 cells were infected with these recombinant strains and the intracellular production of cGMP and cAMP was measured using ELISA kits. Injection of WT ExoY or ExoY[mut2] resulted in a large increase in cGMP levels (Fig. 1h). However, in line with the transfection assay, injection of ExoY[mut0], ExoY[mut1] and ExoY[mut3] effectors did not result in a detectable increase in cellular cGMP (i.e., similar to the uninfected control). Regarding cAMP production in infected cells, only the injection of the ExoY[mut2] or ExoY[mut3] variants, which have a higher adenylyl cyclase catalytic efficiency, produced detectable cAMP (Fig. 1i). Injection by bacteria of WT ExoY, ExoY[mut0] and ExoY[mut1] effectors, did not yield measurable increases in cAMP as compared to uninfected cells (i.e., levels are below the ELISA kit's detection limit specified to be 1.18 pmol/mL according to the manufacturer). We could not measure the intracellular levels of cUMP and cCMP as no specific kits are available for these nucleotides. Instead, we quantified in vitro the nucleotidyl cyclase activity of ExoY variants secreted by the recombinant strains (Supplementary Fig. 2e–h). The results showed the same pattern as those obtained using purified proteins (Fig. 1b–e).

In summary, all these results demonstrate that the amino-acid substitutions introduced into the ExoY nucleotide-binding pocket successfully altered substrate specificity, leading to differential cNMP accumulation in cells. The substrate specificity of the WT and ExoY mutant alleles, as derived from these results and summarized in Fig. 1j, allowed us to further investigate the physiological role of these different cyclic nucleotides in infected cells.

### In NCI-H292 cells, ExoY-derived cGMP dampens the cytotoxicity of ExoT but not that of ExoS

ExoY is generally co-delivered with other T3SS effectors[15]. In ExoS+ strains, it is typically injected together with ExoT and/or ExoS. In contrast to ExoT and ExoS, ExoY does not exhibit detectable cytotoxicity per se, when injected alone at physiological levels by bacteria harboring a single chromosomal copy of the *exoY* gene[29,30,44]. ExoY was even suggested to exert a protective role by attenuating the cytotoxicity of ExoT and ExoS[30,45]. Since ExoY produces diverse cNMPs in cells, we wondered which of these cNMPs could be implicated in the modulation of cytotoxicity induced by ExoT or ExoS. We therefore introduced the various *exoY* alleles into the genome of the *P. aeruginosa* PAO1FΔS(T,Y) and PAO1FΔT(S,Y) strains. These strains were then compared in cytotoxicity assays. As ExoT and ExoS are known to specifically induce cell shrinkage and disruption of the actin cytoskeleton, we quantified cell rounding of infected cells over time[10]. For this purpose, NCI-H292 cells were incubated prior to infection with a red fluorescent probe that labeled the cell cytoplasm, and fluorescence representing the total surface area was monitored using an automated live cell imaging and analysis system. As the ExoY variants were co-injected with a single T3SS effector, either ExoT or ExoS, the cytotoxicity observed could be directly linked to these effectors and

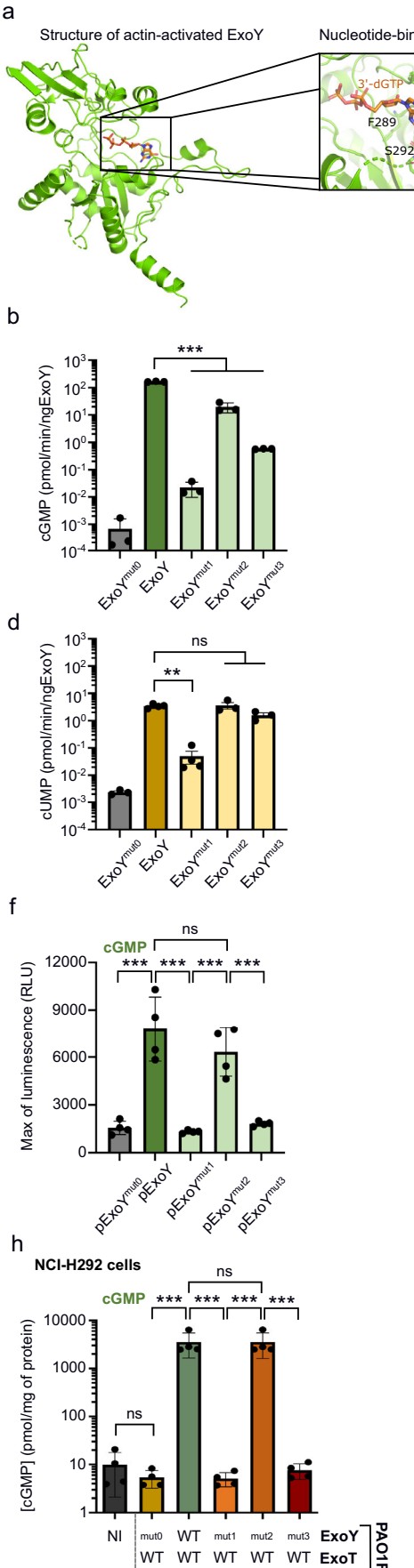

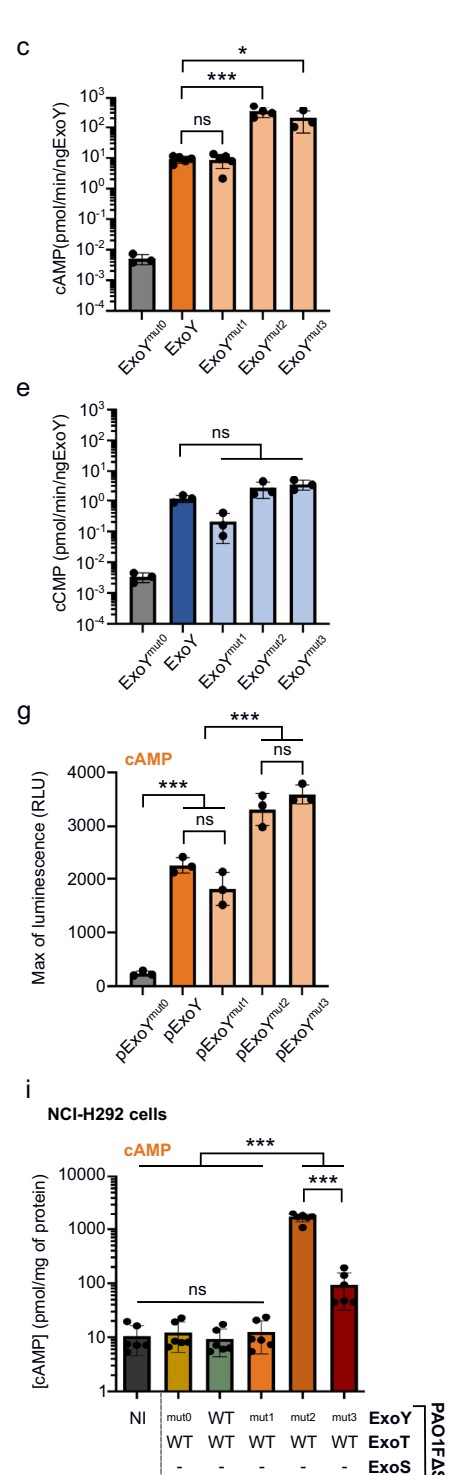

**j**

| ExoY variants | | cNMP activity (pmol/min/ngExoY) | | | |
|---|---|---|---|---|---|
| Variants | Mutations | cGMP | cAMP | cUMP | cCMP |
| ExoY | | 165 | 9,1 | 3,5 | 1,2 |
| ExoY$^{mut0}$ | K81M, K88I | 0 | 0 | 0 | 0 |
| ExoY$^{mut1}$ | N264V, F289V, S292G, D293A | 0,02 | 8,9 | 0,05 | 0,2 |
| ExoY$^{mut2}$ | F83L, S292G | 20 | 343 | 3,7 | 2,6 |
| ExoY$^{mut3}$ | F83L, E258D, S292G | 0,6 | 206 | 1,6 | 3,4 |

**Fig. 1 | Modification of ExoY effector substrate specificity. a** Structure of the *Pseudomonas aeruginosa* ExoY bound to F-actin and 3′-deoxy GTP (3′-dGTP) as solved by *Belyy* et al.[1] with a detailed view of its nucleotide-binding pocket. Modified residues to change ExoY's substrate specificity are highlighted. **b** Analysis of guanylate cyclase (GC) activity, **c** adenylate cyclase (AC), **d** uridylyl cyclase and **e** cytidylyl cyclase activities of purified ExoY proteins. For each condition, we subtracted the value of a blank control, representing the background because no ExoY protein was added, from the raw data. **f** Luminescence activity of the cGMP- or **g** cAMP-GloSensor reporter cell line. Stable cell lines were co-transfected with an expression plasmid encoding an ExoY variant and the Tet-On 3G transactivator expression plasmid. Cells were incubated with 1 μM of Doxycycline for 3 h prior to recording luminescence intensity. The luminescence maximum was extracted and plotted. **h** Quantification of cGMP or **i** cAMP production in NCI-H292 cells by ELISA after 5 h or 3 h of infection, respectively, at MOI 20 with PAO1FΔS(T, Y) strains. **j** Summary table of the catalytic activity of ExoY variants to produce different cNMPs. Values represent the average activity measured using purified proteins (data from Fig. 1b−e). '0' indicates a value below 0.006. The results of plasmid expression in luminescent reporter cells and ELISA assays after infection agree with

the summary table. '-' means that the effector is not expressed due to the deletion of the corresponding gene. NI means 'not infected'. Data of biological replicates represented as mean ± SD, n = 3 (**b**−**e**,**g**), n = 4 (**f**,**h**), n = 6 (**i**). Statistical differences were established by one-way ANOVA followed by Dunnett's multiple comparison test with ExoY as a control group (**b**−**e**), or by Tukey's test (**f**−**i**). Exact P values (**b**): ExoY vs. ExoY$^{mut1}$ = <0.001, ExoY vs. ExoY$^{mut2}$ = <0.001, ExoY vs. ExoY$^{mut3}$ = <0.001, (**c**): ExoY vs. ExoY$^{mut2}$ = <0.001, ExoY vs. ExoY$^{mut3}$ = 0.02, (**d**): ExoY vs. ExoY$^{mut1}$ = 0.001, (**f**): pExoY$^{mut0}$ vs. pExoY = <0.001, pExoY vs. pExoY$^{mut1}$ = <0.001, pExoY$^{mut1}$ vs. pExoY$^{mut2}$ = <0.001, pExoY$^{mut2}$ vs. pExoY$^{mut3}$ = <0.001, (**g**): pExoY$^{mut0}$ vs. pExoY or pExoY$^{mut1}$ = <0.001, pExoY or pExoY$^{mut1}$ vs. pExoY$^{mut2}$ or pExoY$^{mut3}$ = <0.001, (**h**): PAO1FΔS(T,Y$^{mut0}$) vs. PAO1FΔS(T,Y) = < 0.001, PAO1FΔS(T,Y) vs. PAO1FΔS(T,Y$^{mut1}$) = <0.001, PAO1FΔS(T,Y$^{mut1}$) vs. PAO1FΔS(T,Y$^{mut2}$) = <0.001, PAO1FΔS(T,Y$^{mut2}$) vs. PAO1FΔS(T,Y$^{mut3}$) = <0.001, (**i**): NI or PAO1FΔS(T,Y$^{mut0}$) or PAO1FΔS(T,Y) or PAO1FΔS(T,Y$^{mut1}$) vs. PAO1FΔS(T,Y$^{mut2}$) or PAO1FΔS(T,Y$^{mut3}$) = <0.001, PAO1FΔS(T,Y$^{mut2}$) vs. PAO1FΔS(T,Y$^{mut3}$) = <0.001. * p = 0.033, **p = 0.002, ***p < 0.001, n.s., non-significant. Source data are provided as a Source Data file.

potential synergistic effects with the co-injected ExoY variant. Indeed, a strain deleted for ExoS and ExoT but injecting only ExoY did not induce rapid cell retraction as compared to a strain injecting both ExoT and ExoY (Fig. 2a, b). The apparent drop in signal at later time points results from dye dilution and signal loss, not from cell rounding or detachment.

With PAO1FΔS(T,Y) strains, co-injecting ExoT with WT ExoY or ExoY$^{mut2}$, we observed similar kinetic profiles of cell rounding. In contrast, ExoT-induced cell rounding was faster (Fig. 2c) when the toxin was co-injected with the ExoY$^{mut0}$, ExoY$^{mut1}$ or ExoY$^{mut3}$, variants with deficient or significantly diminished cGMP-synthetizing activity (Fig. 1j). Hence, strains expressing an ExoY variant deficient in cGMP production exhibited exacerbated ExoT-mediated cytotoxicity (Fig. 2d). Notably, this effect does not appear to result from variations of ExoT translocation across conditions. (Supplementary Fig. 3a, b).

Similar experiments were carried out with the PAO1FΔT(S,Y) derivatives to analyze potential synergistic effects between ExoS and the ExoY variants. Interestingly, retraction curves for all strains examined were similar, indicating that ExoY did not alter ExoS-mediated cell rounding (Fig. 2e, f). However, the PAO1FΔT(S,Y) derived strains failed to trigger a significant cGMP increase in infected cells as compared to the PAO1FΔS(T,Y) strains co-injecting ExoT with WT ExoY or ExoY$^{mut2}$ (Fig. 2g and Supplementary Fig. 3c). We considered two hypotheses to explain the absence of detectable cGMP (the main cNMP produced by WT ExoY) levels in cells infected with PAO1FΔT(S,Y) strains: i) Feedback inhibition of T3SS effectors injection by ExoS[46,47] that could have prevented ExoY from being efficiently injected, or ii) ExoS interference with the activation of ExoY by F-actin. Both scenarios would indeed result in a lack of cNMP accumulation in infected cells. Western blot (WB) analysis showed that both ExoS and ExoY were well secreted and injected into cells by the different strains (Supplementary Fig. 3a, b), ruling out the first hypothesis. We therefore hypothesize that ExoS might interfere with the activity or activation of ExoY, preventing cNMPs production and thereby limiting any potential modulation of ExoS-induced cytotoxicity. This point is further examined below.

## ExoY-synthesized cGMP specifically alters the downstream effects of ExoT ADPRT activity

ExoT and ExoS are bifunctional toxins with an N-terminal GTPase-activating protein (GAP) domain and a C-terminal ADP-ribosyl-transferase (ADPRT) domain, both of which contribute to cell shrinkage. However, the domain responsible for cell rounding varies across cell types[32,48−52]. To elucidate how ExoT and ExoS mediate cell rounding in our cell model, we investigated the contributions of their GAP and ADPRT activities. Point mutations were introduced into the

genomes of PAO1FΔS(T,Y) or PAO1FΔT(S,Y) strains to selectively inactivate each activity[49−51,53−59] (Fig. 3a). Cell rounding was then evaluated in cells infected with the mutant strains. Our results showed that the ADPRT activity of both toxins primarily drives cell retraction in NCI-H292 cells (Fig. 3b, c and Supplementary Fig. 4a, b). The data further indicated that the GAP activity of ExoT partly counteracted the cell rounding induced by the ADPRT domain of ExoT, as the total cell surface area was significantly lower after 8 h of infection with ExoT$^{GAP-}$ mutant as compared to WT ExoT. Given that ExoY-derived cGMP attenuates ExoT cytotoxicity, we therefore hypothesized that in NCI-H292 cells, ExoY-synthesized cGMP specifically interferes with ExoT ADPRT activity and/or its downstream effects, thereby protecting cells from rounding.

To test this hypothesis, we generated a chimeric ExoT variant by replacing the ExoT ADPRT domain with that of ExoS (Fig. 3d). The variant was introduced into PAO1FΔS(T,Y) strains via allelic exchange, and cytotoxicity assays were performed on cells infected with *P. aeruginosa* strains injecting either WT or chimeric ExoT variants. As shown above, the ExoT-induced cell rounding was dampened by the co-injection of the WT ExoY as compared to that observed upon co-injection of the ExoY variants unable to produce cGMP (ExoY$^{mut0}$ and ExoY$^{mut1}$) (Fig. 3e, f). In contrast, the cell retraction induced by the chimeric ExoT$^{ADPRT\_ExoS}$ was not affected by the ability of co-injected ExoY variants to generate cGMP. Furthermore, the cell retraction caused by the chimeric ExoT$^{ADPRT-ExoS}$ was comparable to that induced by ExoS, confirming that the chimera is functional in cells. These results demonstrate that cGMP synthesized by ExoY specifically interferes with the signaling pathway targeted by ExoT's ADPRT domain, thereby attenuating ExoT's disruptive impact on the host cell cytoskeleton. These results also highlight that the ExoS ADPRT activity is either insensitive to ExoY cGMP production or, more likely, that ExoS ADPRT activity counteracts ExoY activity as illustrated by the reduced accumulation of cGMP when both ExoS and ExoY were injected (Fig. 2g and Supplementary Fig. 3c).

## In NCI-H292 cells, ExoS ADPRT activity restricts ExoY activation

We infected the luminescent cGMP reporter cell line, representing intracellular cGMP production, with PAO1FΔS(T,Y) or PAO1FΔT(S,Y) strains to test the hypothesis that, compared with ExoT, ExoS activity interferes with ExoY activation and thus with its production of cGMP. Luminescence intensities after infection of the reporter cell line with recombinant PAO1FΔS(T,Y) strains co-injecting ExoY with ExoT or ExoT variants were all similar to those obtained when ExoY was injected alone, indicating that, as expected, ExoT activities do not prevent ExoY from being active (Fig. 3g). In contrast, co-injection of active ExoS toxin with ExoY decreased the intensity of luminescence

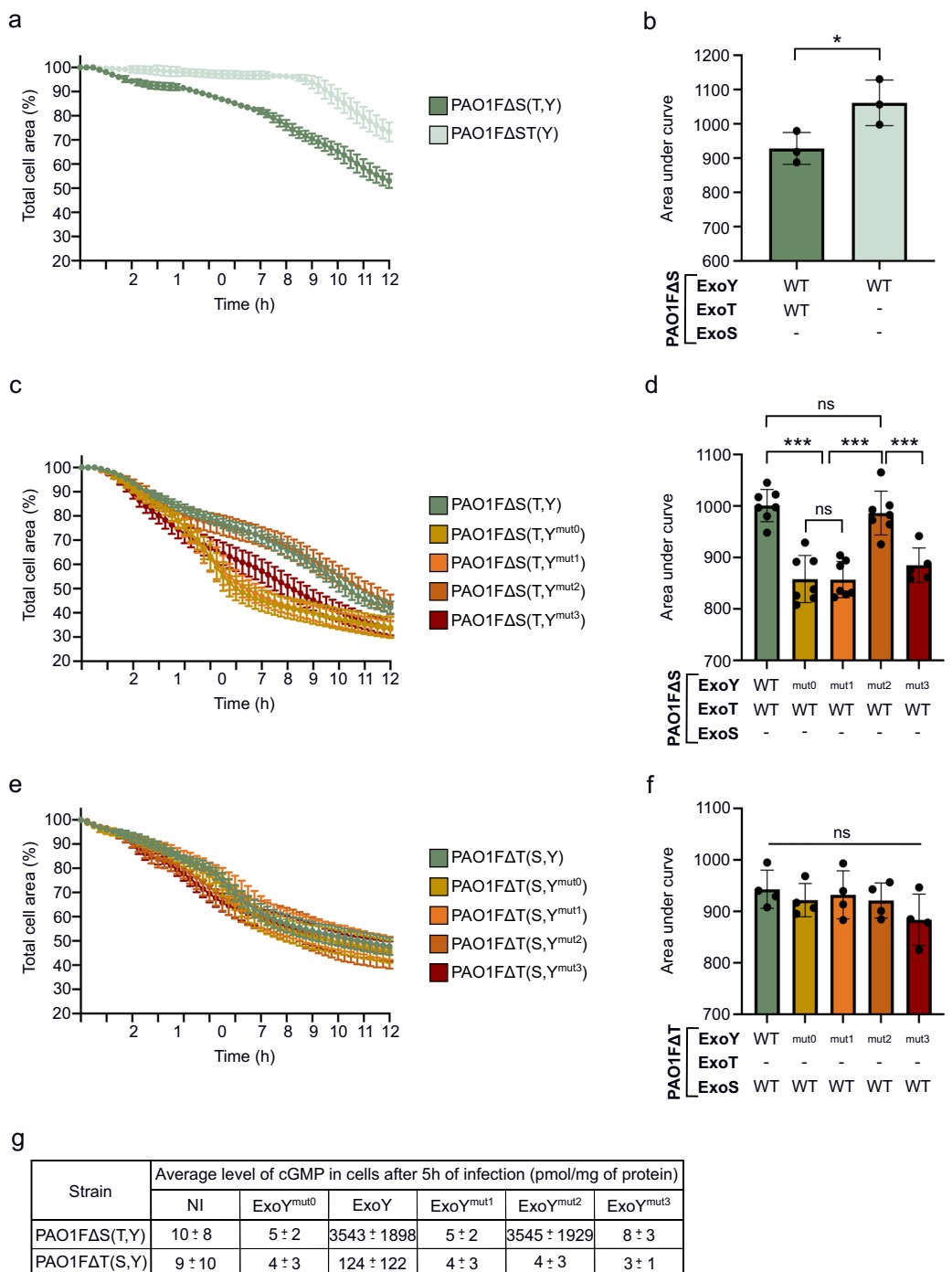

g

| Strain | Average level of cGMP in cells after 5h of infection (pmol/mg of protein) | | | | | |
|---|---|---|---|---|---|---|
| | NI | ExoY$^{mut0}$ | ExoY | ExoY$^{mut1}$ | ExoY$^{mut2}$ | ExoY$^{mut3}$ |
| PAO1FΔS(T,Y) | 10 ± 8 | 5 ± 2 | 3543 ± 1898 | 5 ± 2 | 3545 ± 1929 | 8 ± 3 |
| PAO1FΔT(S,Y) | 9 ± 10 | 4 ± 3 | 124 ± 122 | 4 ± 3 | 4 ± 3 | 3 ± 1 |

corresponding to cGMP production compared with injection of ExoY alone (Fig. 3h). Only co-injection of an ExoS variant with inactive ADPRT activity completely restored cGMP production by ExoY in NCI-H292 cells, validating the above proposition that ExoS ADPRT activity impedes ExoY activity. The lack of ExoY activity could result from a local inhibition of actin polymerization by the ADPRT activity of ExoS, thus disrupting the activation of ExoY. As a result, compared to ExoT, ExoS activity prevents the toxin from a possible feedback inhibition by ExoY-synthesized cGMP. Interestingly, when the three effectors are co-injected, bacterial virulence seems to be primarily induced by ExoS, as shown by the comparable levels of cell retractions between the strains PAO1F(S,T,Y) and PAO1FΔT(S,Y), which are significantly different from the PAO1FΔS(T,Y) strain (Fig. 3i, j). In this context, the accumulation of cGMP by ExoY at levels sufficient to counteract ExoT-mediated

cytotoxicity is markedly reduced (Fig. 3k), corroborating our previous conclusion that ExoS is able to suppress the activity of ExoY in NCI-H292 cells. These findings further highlight the complex interplay among the three co-injected effectors and indicate that, although ExoY can modulate ExoT-induced cytotoxicity, its impact remains moderate in the presence of ExoS.

## CrkII is the downstream target of ExoT's ADPRT activity modulated by ExoY-induced cGMP

The GAP domains of ExoS and ExoT target the same Rho family GTPases, including Rho, Rac1, and Cdc42[48,50,53,60], while their ADPRT domains target distinct cellular proteins. ExoS ADP-ribosylates various proteins, such as Ras, Rac1 and the ERM family[49,61–66], whereas ExoT ADP-ribosylates Crk family proteins and actin[67–70]. Since cGMP

**Fig. 2 | Production of cGMP by ExoY alters ExoT cytotoxicity but not ExoS cytotoxicity in NCI-H292 cells. a** Cell retraction over time in infected NCI-H292 cells. Cells were infected with PAO1FΔS(T,Y) strains injecting ExoY with or without ExoT. Intoxication was quantified every 15 min by time-lapse microscopy. One biological replicate represented as a mean ± SD. **b** Cytotoxicity level of PAO1FΔS(T,Y) strains. The area under the curve (AUC) for each infection curve was plotted. Lower AUCs correspond to higher cytotoxicity. Each dot represents a separate cell retraction assay (biological replicate). **c** ExoT-induced cell retraction over time in NCI-H292 cells. Cytotoxicity assay was similar to (**a**), but cells were infected with PAO1FΔS(T,Y) strains co-injecting ExoT with an ExoY variant. **d** Cytotoxicity level of ExoT-expressing strains. Representation is similar to (**b**) but using the infection assays with PAO1FΔS(T,Y) strains co-injecting ExoT with an ExoY variant. **e** ExoS-induced cell retraction over time in NCI-H292 cells. Cytotoxicity assay was similar to (**a**) and (**c**), but cells were infected with PAO1FΔT(S,Y) strains co-injecting ExoS with an ExoY variant. **f** Cytotoxicity level of ExoS-expressing strains. Representation is similar to (**b**) and (**d**) but using the infection assays with PAO1FΔT(S,Y) strains. **g** Comparative table of intracellular cGMP production after infection of NCI-H292 cells with PAO1FΔS(T,Y) or PAO1FΔT(S,Y) strains. For both conditions, cells were infected for 5 h at MOI 20. Intracellular cGMP production was quantified using ELISA kit. Values obtained for PAO1FΔS(T,Y) strains have been extracted from Fig. 1h. The mean cGMP accumulation of each sample is shown, together with the SD. '-' means that the effector is not expressed due to deletion of the corresponding gene. NI means 'not infected'. Data of biological replicates represented as mean ± SD, n = 3 (**b**), n = 4 (**f**), n = 5 (**d**). Statistical differences were established by an unpaired two-tailed t-test (**b**), or by one-way ANOVA followed by Tukey's multiple comparison test (**d**, **f**). Exact P values (**b**): PAO1FΔS vs. PAO1FΔST = 0.05 (t = 2.84; df=4 with 95% confidence intervals between 3.15 and 263), (**d**): PAO1FΔS(T,Y) vs. PAO1FΔS(T,Y$^{mut0}$) or PAO1FΔS(T,Y$^{mut1}$) = <0.001, PAO1FΔS(T,Y$^{mut0}$) or PAO1FΔS(T,Y$^{mut1}$) vs. PAO1FΔS(T,Y$^{mut2}$) = <0.001, PAO1FΔS(T,Y$^{mut2}$) vs. PAO1FΔS(T,Y$^{mut3}$) = <0.001. * p = 0.033, **p = 0.002, ***p < 0.001, n.s., non-significant. Source data are provided as a Source Data file.

production by ExoY only altered ExoT's ADPRT activity, we reasoned that this effect might depend primarily on the eukaryotic targets of ExoT's ADPRT, namely the Crk proteins.

We therefore evaluated the possibility that Crk family proteins are involved in the cGMP-antagonized ExoT cytotoxicity, by generating Crk$^{-/-}$ NCI-H292 cells using CRISPR-Cas9 technology. For this purpose, a gRNA was designed to specifically target the *crk* gene and not the *crkL* gene (Supplementary Fig. 5a). The *crk* gene encodes two spliced products, CrkI and CrkII, which differ in size. CrkII has an additional SH3 domain at its C-terminus, resulting in a 40 kDa protein, whereas CrkI is a 28 kDa protein (Fig. 4a). The *crkL* gene encodes a paralogous Crk-like protein (CrkL) that shares high sequence homology with CrkII. All three proteins are considered adaptor proteins and play an essential role in integrating signals from a wide variety of cellular processes[71,72]. However, since ExoT ADP-ribosylation has been described to modify CrkI/II proteins and not CrkL, we focused our analysis on CrkI/II proteins.

A clonal population was selected, in which four bases were deleted from the beginning of the coding sequence of the *crk* alleles, theoretically ablating the expression of WT CrkI/II proteins (Supplementary Fig. 5a). Both CrkI and CrkII production were indeed knocked-out in deficient cells compared to WT cells (Supplementary Fig. 5b). Low-intensity bands of the same size as CrkI and CrkII were nevertheless detected in deficient cells. This suggests that the obtained clones were not fully depleted of CrkI/CrkII, possibly because complete depletion could be detrimental to cell physiology. WT and Crk-deficient cells were then infected with PAO1FΔS(T,Y) strains and cell rounding was recorded as previously described. In contrast to WT cells, Crk$^{-/-}$ cells showed a reduced ExoT-induced cell rounding, represented by an increase in the area under the curve (AUC), with no significant difference between strains regardless of the co-injected ExoY effector variant (Fig. 4b and Supplementary Fig. 6a–d). These data indicate that CrkI and/or CrkII are the downstream targets of the ExoT/ADPRT activity blunted by ExoY-synthesized cGMP. To identify the Crk isoform involved, Crk-deficient cells were complemented with a lentiviral plasmid encoding either CrkI or CrkII. Rescuing CrkII expression in Crk-deficient cells resulted in infection profiles with PAO1FΔS(T,Y) strains similar to those observed in wild-type cells, whereas CrkI expression did not restore similar profiles (Fig. 4b, c and Supplementary Fig. 6e–h). Crk$^{-/-}$::CrkII cells were indeed as sensitive as WT cells to cGMP-dependent ExoT cytotoxicity. These results confirmed that CrkII is the main target of the cytotoxicity of ExoT modulated by ExoY during the infection of the cells.

### ExoY-synthesized cGMP dampens CrkII dephosphorylation induced by ExoT's ADPRT activity

CrkII is a part of focal adhesion (FA) complexes composed of kinases (Src and FAK), docking molecules (p130Cas, paxillin) and downstream proteins such as DOCK180 and C3G. Upon stimulation, CrkII is tyrosine-phosphorylated at Y221 by activated Abelson's tyrosine-protein kinase (Abl)[73–76], and translocated to the membrane[74,77–79] where the adaptor protein CrkII is dephosphorylated to interact with tyrosine-phosphorylated p130Cas[78,80–86] and other partners, leading to actin cytoskeleton reorganization, cell migration and cell spreading[87]. However, Crk signal transduction pathways are disrupted after injection of ExoT by *P. aeruginosa*. Abl/CrkII interaction and CrkII phosphorylation[74,88] are inhibited by ExoT through ADP-ribosylation of cytoplasmic CrkII at Arg20[67], thus preventing CrkII membrane localization. In addition, CrkII interaction with upstream signaling molecules such as activated p130Cas is also inhibited by ExoT ADP-ribosylation[70]. Therefore, ExoT's ADPRT activity induces loss of cell adhesion and cell-matrix interaction, leading to cell rounding and eventually to a programmed apoptotic cell death called anoikis[70,89,90].

We therefore explored whether the ExoY-synthesized cGMP could limit the ExoT-induced CrkII dephosphorylation at Y221. CrkI, CrkII, and phospho-CrkII (at Y221) levels were analyzed by WB after infection of cells with the different PAO1FΔS(T,Y) strains (Fig. 4d). CrkI and CrkII levels remained constant after four hours of infection, while phospho-CrkII levels were drastically reduced in infected cells as compared to the uninfected condition but were similar regardless of the PAO1FΔS(T,Y) strain used for infection. Furthermore, the level of phospho-CrkII was similar between the uninfected condition and the conditions in which cells were infected for four hours with the strains PAO1FΔST(Y) (ExoY injection only) and PAO1FΔSTY (injecting none of the T3SS effectors) (Fig. 4d). These results indicate that: (i) ExoT cytotoxicity leads to CrkII dephosphorylation, as previously described[74,88], and (ii) injection of active WT ExoY alone does not alter phospho-CrkII level per se. It should be noted that in NCI-H292 cells, prior to infection, CrkII is highly phosphorylated. Infection was therefore repeated at different times to follow kinetics of CrkII dephosphorylation by ExoT activity in order to visualize potential differences between the strains related to the distinct NC activities of the injected ExoY variants. As shown in Fig. 4e, phospho-CrkII levels were not altered over time when cells were infected with the PAO1FΔST(Y) strain, injecting only active ExoY, demonstrating that cGMP production by ExoY is not directly involved in CrkII phosphorylation nor in altering phospho-CrkII levels. However, as expected, CrkII is dephosphorylated over time due to the ExoT activity co-injected with ExoY. Interestingly, when ExoT is co-injected with ExoY variants defective in cGMP production (ExoY$^{mut0}$, ExoY$^{mut1}$ or ExoY$^{mut3}$), the rate of CrkII dephosphorylation by ExoT activity is increased, particularly after 2 h of infection (Fig. 4e, f). These data demonstrated that the cGMP produced by ExoY attenuated the ExoT-dependent CrkII dephosphorylation at Y221 in infected NCI-H292 cells, thereby decreasing ExoT's ADPRT-mediated cytotoxicity (Fig. 2c, d).

To further assess whether cGMP contributes to the modulation of the ExoT-induced CrkII dephosphorylation, the infection medium

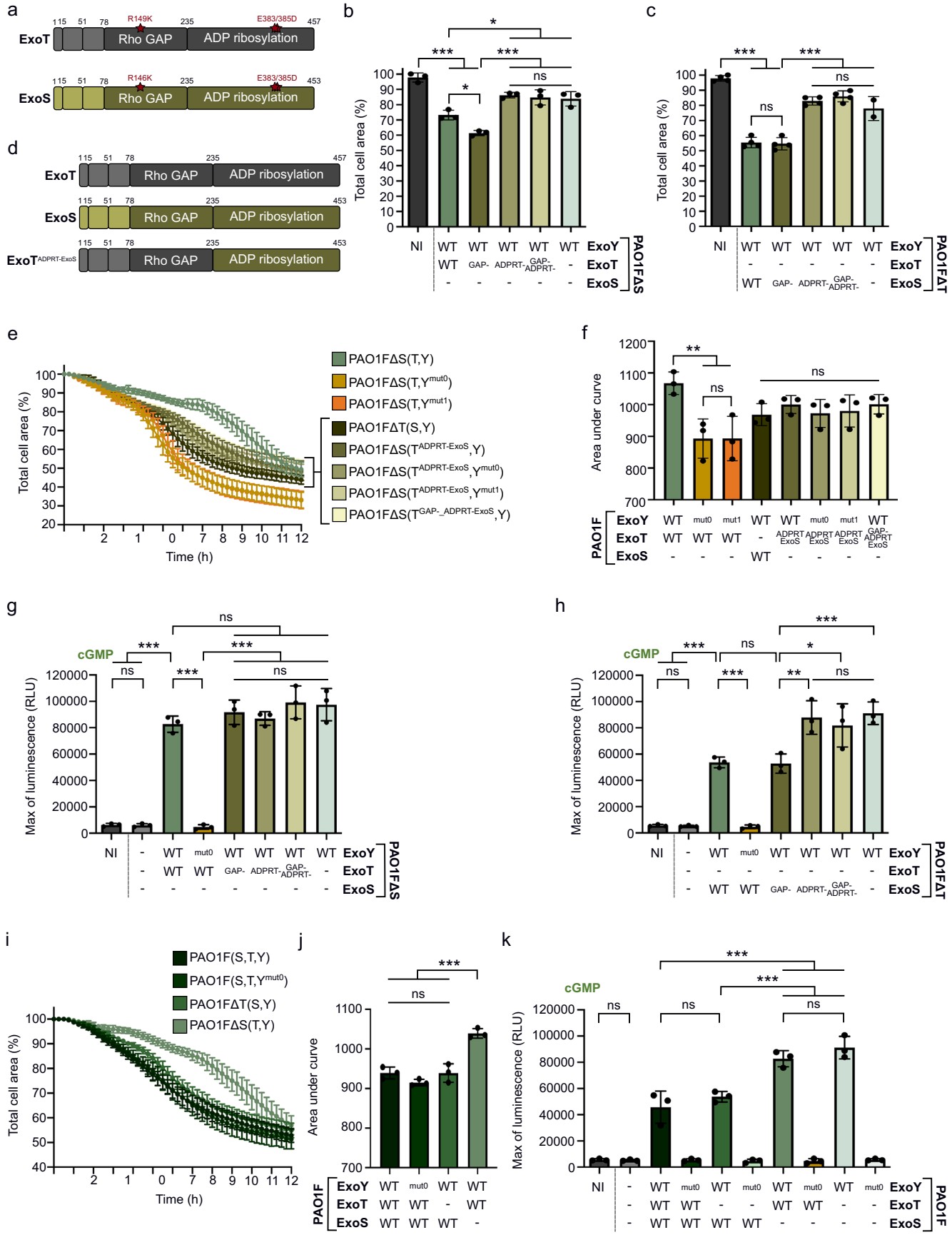

**Fig. 3 | The antagonistic effect of ExoY cGMP production on ExoT cytotoxicity depends on ExoT's ADPRT activity in NCI-H292 cells. a** Substitutions performed on *exoT* and *exoS* genes to inactivate their GAP and/or ADPRT activity. **b** Quantification of cell retraction induced by ExoT toxin or its derivatives. Cells were infected at MOI 50 for 8 h. Values indicate the percentage of total cell surface after infection compared with the surface area before infection (T 0 h set to 100%). Each data point represents a biological replicate corresponding to the average value of 16 images (technical replicates). **c** Quantification of cell retraction induced by ExoS toxin or its derivatives. Representation is similar to (**b**), but cells were infected at MOI 20. **d** Representation of ExoT, ExoS and modified ExoT toxins with their respective GAP and ADPRT domains. In the modified ExoT toxin (chimeric toxin), the ADPRT domain of ExoT has been replaced by that of ExoS to form the ExoT$^{ADPRT-ExoS}$ toxin. **e** Infection assay of NCI-H292 cells with PAO1FΔS(T,Y) strains co-injecting WT ExoT or ExoT$^{ADPRT-ExoS}$ with different ExoY variants (WT ExoY, ExoY$^{mut0}$ and ExoY$^{mut1}$). One biological replicate represented as a mean ± SD. **f** Cytotoxicity level of strains expressing WT ExoT or modified toxin. The area under the curve (AUC) for each infection curve was plotted on a bar graph. Each dot represents a separate cell retraction assay (biological replicate). **g** Luminescence activity of the cGMP-GloSensor reporter cell line. Cells were infected for 2 h at MOI 20 with PAO1FΔS(T,Y) strains co-injecting ExoY with WT ExoT or its inactive mutants prior recording luminescence intensity. The luminescence maximum was extracted and plotted. Each dot represents a biological replicate. **h** Similar to (**g**) but cells were infected with PAO1FΔT(S,Y) strains co-injecting ExoY with WT ExoS or its inactive mutants. **i** Cell retraction over time in infected NCI-H292 cells. Cells were infected with PAO1F(S,T,Y) strains at MOI 20 and cell retraction was quantified every 15 min by time-lapse microscopy. One biological replicate represented as a mean ± SD. **j** Cytotoxicity levels of PAO1F(S,T,Y) strains. The area under the curve

(AUC) for each infection curve was plotted. Each dot represents a separate cell retraction assay (biological replicate). **k** Luminescence activity of the cGMP-GloSensor reporter cell line infected for 2 h at MOI 20 with PAO1F(S,T,Y) strains. Luminescence intensity was recorded every 3 min for 5 h and the luminescence maximum was extracted and plotted. Each dot represents a biological replicate. '-' means that the effector is not expressed due to deletion of the corresponding gene. NI means 'not infected'. Data of biological replicates represented as mean ± SD, n = 3 (**b**, **f–h**, **j**, **k**), n = 4 (**c**). Statistical differences were established by one-way ANOVA followed by Tukey's multiple comparison test. Exact P values (**b**): NI vs. PAO1FΔS(T,Y) or PAO1FΔS(T$^{GAP-}$,Y) or = < 0.001, PAO1FΔS(T,Y) vs. PAO1FΔS(T$^{GAP-}$,Y) = 0.01, PAO1FΔS(T,Y) vs. PAO1FΔS(T$^{ADPRT-}$,Y) = 0.006, PAO1FΔS(T,Y) vs. PAO1FΔS(T$^{GAP-/ADPRT-}$,Y) = 0.02, PAO1FΔS(T,Y) vs. PAO1FΔST(Y) = 0.03, PAO1FΔS(T$^{GAP-}$,Y) vs. PAO1FΔS(T$^{ADPRT-}$,Y) or PAO1FΔS(T$^{GAP-/ADPRT-}$,Y) or PAO1FΔST(Y) = < 0.001, (**c**): NI vs. PAO1FΔT(S,Y) or PAO1FΔT(S$^{GAP-}$,Y) = < 0.001, PAO1FΔT(S$^{GAP-}$,Y) vs. PAO1FΔT(S$^{ADPRT-}$,Y) or PAO1FΔT(S$^{GAP-/ADPRT-}$,Y) or PAO1FΔST(Y) = < 0.001, (**f**): PAO1FΔS(T,Y) vs. PAO1FΔS(T,Y$^{mut0}$) or PAO1FΔS(T,Y$^{mut1}$) = 0,006, (**g**): PAO1FΔS(T,Y) vs. NI or PAO1FΔSTY = < 0.001, PAO1FΔS(T,Y) vs. PAO1FΔS(T,Y$^{mut0}$) = <0.001, PAO1FΔS(T,Y$^{mut0}$) vs. PAO1FΔS(T$^{GAP-}$,Y) or PAO1FΔS(T$^{ADPRT-}$,Y) or PAO1FΔS(T$^{GAP-/ADPRT-}$,Y) or PAO1FΔST(Y) = < 0.001, (**h**): PAO1FΔT(S,Y) vs. NI or PAO1FΔSTY = < 0.001, PAO1FΔT(S,Y) vs. PAO1FΔT(S,Y$^{mut0}$) = <0.001, PAO1FΔT(S$^{GAP-}$,Y) vs. PAO1FΔT(S$^{ADPRT-}$,Y) = 0.002, PAO1FΔT(S$^{GAP-}$,Y) vs. PAO1FΔT(S$^{GAP-/ADPRT-}$,Y) = 0.01, PAO1FΔT(S$^{GAP-}$,Y) vs. PAO1FΔST(Y) = < 0.001, (**j**): PAO1FΔS(T,Y) vs. PAO1F(S,T,Y) or PAO1F(S,T,Y$^{mut0}$) or PAO1FΔT(S,Y) = < 0.001, (**k**): PAO1F(S,T,Y) vs. PAO1FΔS(T,Y) or PAO1FΔS(T,Y$^{mut0}$) or PAO1FΔST(Y) = < 0.001, PAO1FΔT(S,Y) vs. PAO1FΔS(T,Y) or PAO1FΔS(T,Y$^{mut0}$) or PAO1FΔST(Y) = < 0.001. *p = 0.033, **p = 0.002, ***p < 0.001, n.s., non-significant. Source data are provided as a Source Data file.

was supplemented with compounds known to stimulate cGMP production in cells, and phospho-CrkII levels were analyzed by WB during infection. Spermine NONOate is a nitric oxide donor that activates soluble guanylate cyclases (GCs) and thus increases intracellular cGMP levels[91], while 8-Br-cGMP is a cell-permeable cGMP analog that is more resistant to hydrolysis by phosphodiesterases than cGMP. When cells were infected with the strain co-injecting ExoT with ExoY$^{mut0}$ (unable to produce all cNMPs), the addition of either compound limited CrkII dephosphorylation induced by ExoT activity over time compared with the untreated condition (Fig. 4g, h). These data are consistent with the idea that cGMP produced by ExoY contributes to delay ExoT-induced CrkII dephosphorylation and, consequently, to limit ExoT-mediated cell rounding.

Overall, the results highlight the complex interaction between ExoY and ExoT effectors and their respective activities in infected cells, as illustrated in Fig. 5.

**Inhibition of ExoT cytotoxicity by ExoY depends on cell model**
To reinforce and generalize our discovery on the interaction between ExoT and ExoY, we examined their effects on other epithelial cell lines commonly used in studies of *P. aeruginosa* infection, HeLa cells[50,52] and A549 alveolar epithelial cells[13,18,47].

Co-injection of ExoT with ExoY$^{mut0}$ or ExoY$^{mut1}$ resulted in an increase in ExoT-induced cell retraction in HeLa cells compared to WT ExoY (Fig. 6a, b), similarly to what was observed with NCI-H292 cells (Fig. 2c, d), although the overall effect was less pronounced. As similar levels of ExoT were injected by these strains (Fig. 6c) we concluded that, as in NCI-H292 cells, the guanylate cyclase negative variants ExoY$^{mut0}$ and ExoY$^{mut1}$ could not inhibit ExoT cytotoxicity. However, in contrast to NCI-H292 cells where only the ADPRT activity of ExoT is involved in cell retraction, we found that in HeLa cells, the GAP activity of ExoT also plays an important role in cell retraction (Fig. 6d). So, since ExoY-synthesized cGMP only dampens ExoT's ADPRT activity, its effect on ExoT cytotoxicity is weaker in HeLa cells, in contrast to NCI-H292 cells, where ExoT cytotoxicity is entirely ADPRT-dependent. Another interesting observation is that the HeLa cell rounding was drastically increased upon infection with strains co-

injecting ExoT with ExoY$^{mut2}$ or ExoY$^{mut3}$ (Fig. 6a, b). This increase in strain virulence is likely due to the significant over-injection (up to three-fold) of ExoT into cells when co-injected with these two ExoY mutants, as revealed by WB (Fig. 6c). Whether the higher amount of ExoT in HeLa cells is linked to the higher AC activity of ExoY$^{mut2}$ and ExoY$^{mut3}$ and the putative mechanisms involved, or a greater effector stability remains to be clarified.

Similar *P. aeruginosa* infection experiments were carried out on A549 alveolar epithelial cells. In contrast to NCI-H292 and HeLa cells, the co-injection of ExoY did not counteract ExoT cytotoxicity in A549 cells, as the ExoT-induced cell retraction curve was indistinguishable from those obtained with the defective GC variants, ExoY$^{mut0}$ and ExoY$^{mut1}$ (Fig. 7a, b). WB confirmed that similar levels of ExoT were injected by all three strains (Fig. 7c). However, only very low levels of cGMP (and cAMP) were measured in A549 cells infected with the PAO1FΔS(T,Y) strain, indicating a severe defect in ExoY activation in these cells at variance with what was seen in NCI-H292 cells (Fig. 7d). Additionally, the A549 cell rounding seems to depend solely on the GAP activity of ExoT and not on its ADPRT activity, in contrast to NCI-H292 cells (Fig. 7e). Furthermore, given that CrkII was the key component mediating the synergy between ExoT and ExoY effectors in NCI-H292 cells, we examined CrkI and CrkII expression levels in A549 cells by WB. While CrkI levels were similar between cell lines (Fig. 7f), CrkII levels were very low in A549 cells, comparable to those found in Crk-deficient NCI-H292 cells (Supplementary Fig. 5b). This lack of CrkII expression likely explains why ExoT-induced cell rounding in A549 cells is entirely driven by the GAP activity of ExoT rather than its ADPRT activity. Moreover, as with HeLa cells, ExoT-induced cell rounding was markedly enhanced when A549 cells were infected with strains co-injecting ExoT with ExoY$^{mut2}$ or ExoY$^{mut3}$ (Fig. 7a, b). This increase in ExoT cytotoxicity was again likely due to the higher amounts of ExoT injected into A549 cells by these two PAO1FΔS(T,Y) strain derivatives (Fig. 7c).

Overall, the comparison of *P. aeruginosa* infection outcome in the three different cell lines showed that ExoY cNMP production affects ExoT cytotoxicity in a complex, cell-dependent manner, influenced by ExoT's activity involved in cell rounding and by CrkII expression. The downstream effects of ExoT's ADPRT activity are tempered by ExoY's

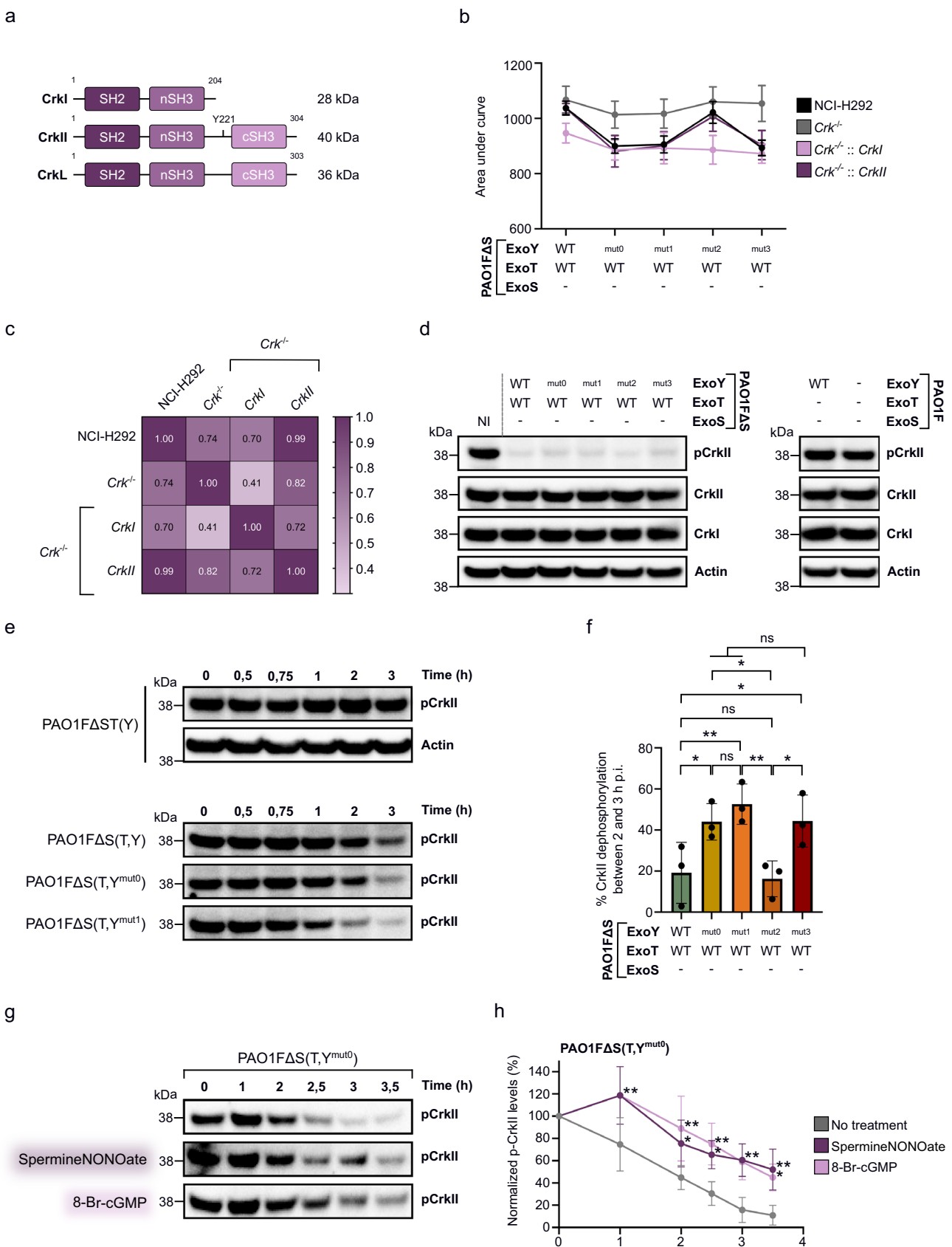

**Fig. 4 | ExoY cGMP production limits ExoT-induced CrkII dephosphorylation in NCI-H292 cells. a** Representation of CrkI, CrkII, and CrkL proteins with their respective Src homology 2 and 3 domains (SH2 and SH3, respectively). The phosphorylation site on CrkII (Y221) is indicated. **b** Infection profile of NCI-H292 cells and Crk-deficient or complemented cells. Cell retractions were recorded after 12 h of infection and the AUC of each cell retraction curve was plotted on the bar chart. For each cell line, the dots were connected to represent the infection profile. Each point represents the average of five biological replicates. Error bars indicate SD. **c** Person correlation between infection profiles of each cell lines from experiment (**b**). **d** Western blot (WB) analysis showing levels of CrkI, CrkII and phosphorylated CrkII (Y221) in infected NCI-H292 cells. Cells were infected with the strains for 4 h. PAO1FΔST(Y) and PAO1FΔSTY strains, injecting only ExoY or no T3SS effector respectively, were also used as controls. **e** WB analysis of phosphorylated CrkII protein (at Y221) over time in infected cells. NCI-H292 cells were infected for the indicated time with strains injecting or not ExoT together with an ExoY variant. **f** CrkII dephosphorylation observed between 2- and 3-hours post-infection. Phospho-CrkII (Y221, p-CrkII) levels were normalized to actin levels (loading control) and p-CrkII before infection (Time 0 h) was set to 100% to determine the percentage of p-CrkII remaining at 2 and 3 h post infection. Each dot represents a biological replicate. **g** WB analysis of phosphorylated CrkII protein over time in infected NCI-H292 cells supplemented or not with compounds that stimulate cGMP production.

SpermineNONOate (50 µM) or 8-Br-cGMP (500 µM) were added in infection medium. **h** p-CrkII levels over time after infection with PAO1FΔS(T,$Y^{mut0}$) strain. The pCrkII level for each infection time was normalized by actin and compared to the uninfected condition (set at 100%). Each dot represents a biological replicate. '-' means that the effector is not expressed due to deletion of the corresponding gene. NI means 'not infected'. Data of biological replicates represented as mean ± SD, n = 5 (**b**), n = 3 (**f, h**). Statistical differences were established by one way ANOVA followed by uncorrected Fisher's LSD test (**f**) or by two ways ANOVA followed by Dunnett's multiple comparison test with no treatment as the control group (**h**). Exact P values (**f**): PAO1FΔS(T,Y) vs. PAO1FΔS(T,$Y^{mut0}$) = 0.02, PAO1FΔS(T,Y) vs. PAO1FΔS(T,$Y^{mut1}$) = 0.004, PAO1FΔS(T,Y) vs. PAO1FΔS(T,$Y^{mut3}$) = 0.02, PAO1FΔS(T,$Y^{mut0}$) vs. PAO1FΔS(T,$Y^{mut2}$) = 0. 01, PAO1FΔS(T,$Y^{mut1}$) vs. PAO1FΔS(T,$Y^{mut2}$) = 0. 003, PAO1FΔS(T,$Y^{mut2}$) vs. PAO1FΔS(T,$Y^{mut3}$) = 0. 01, (**h**): no treatment vs. SpermineNONOate or 8-Br-cGMP (1 h) = 0.002, no treatment vs. SpermineNONOate (2 h) = 0.04, no treatment vs. 8-Br-cGMP (2 h) = 0.002, no treatment vs. SpermineNONOate (2.5 h) = 0.01, no treatment vs. 8-Br-cGMP (2.5 h) = 0.002, no treatment vs. SpermineNONOate (3 h) = 0.002, no treatment vs. 8-Br-cGMP (3 h) = 0.003, no treatment vs. SpermineNONOate (3.5 h) = 0.004, no treatment vs. 8-Br-cGMP (3.5 h) = 0.02. * p = 0.033, **p = 0.002, ***p < 0.001, n.s., non-significant. Source data are provided as a Source Data file.

production of cGMP, while the downstream effects of ExoT's GAP activity are unaffected, possibly because it may prevent ExoY activation.

## Discussion

ExoY is a member of the bacterial nucleotidyl cyclase (NC) family, alongside CyaA from *Bordetella pertussis* and Edema Factor (EF) from *Bacillus anthracis*, which manipulate host cNMP signaling pathways to enhance pathogen survival and proliferation. However, instead of producing supraphysiological levels of cAMP like CyaA and EF, ExoY generates large amounts of cGMP. To our knowledge, no specific function has been ascribed to the cGMP produced by ExoY during infection. Our results demonstrate that in NCI-H292 cells, cGMP production by ExoY attenuates the cytotoxicity of ExoT, the only T3SS effector conserved in all T3SS-expressing *P. aeruginosa* strains. When ExoY variants defective in cGMP production were co-injected with ExoT, ExoT-induced cell retraction was exacerbated. On the contrary, ExoS cytotoxicity, was not inhibited by ExoY in this cell line. The reduced cGMP generation observed under these conditions likely explains the absence of an ExoY-mediated effect. We hypothesize that ExoS, via its ADPRT activity, can locally disrupt the actin polymerization, thereby suppressing the F-actin-dependent ExoY activation and consequently cGMP synthesis. We therefore further examined how ExoY modulates ExoT cytotoxicity in NCI-H292 cells. We focused on the ADPRT domain of ExoT and its targets, the Crk family proteins, as ExoT-mediated cell retraction is driven by ExoT's ADPRT activity. In the NCI-H292 cell model, we found that when ExoY and ExoT are co-injected, ExoY-derived cGMP limits the ExoT-mediated dephosphorylation of CrkII at Y221, which occurs as a result of ADP-ribosylation. As a consequence, by limiting CrkII-Y221 dephosphorylation, ExoY-synthetized cGMP directly attenuates ExoT cytotoxicity. Conversely, ExoY variants unable to produce cGMP had no effect on ExoT-mediated CrkII dephosphorylation and thus on ExoT cytotoxicity, leading to rapid cell retraction by disruption of focal adhesion sites.

The molecular mechanisms that link the ADP-ribosylation of CrkII by ExoT and its Y221 dephosphorylation remain to be precisely elucidated, as well as how the cGMP produced by ExoY can interfere with this process. It has been described that (i) CrkII is specifically dephosphorylated by PTP1B[86] and that (ii) PTP1B phosphatase activity can be indirectly regulated by cGMP. Indeed, PTP1B is regulated by EGFR[92,93] whose activation is controlled by cGMP-dependent protein kinases (PKGI and PKGII)[94–97]. Activation of these kinases by cGMP may thus inhibit EGFR activation and, consequently, reduce PTP1B activity. We

therefore propose that during infection of NCI-H292 cells by *P. aeruginosa*, two mechanisms occur: Firstly, the ADPRT activity of ExoT limits the production of phosphorylated CrkII by preventing the interaction between CrkII and Abl[74,88], and secondly, the phosphatase activity of PTP1B further reduces the level of previously phosphorylated CrkII. Together, these mechanisms lead to rapid dephosphorylation of CrkII at Y221 and consequently induce cell rounding through loss of cell adhesion and cell-matrix interaction. However, high cGMP levels generated by activated ExoY could inhibit PTP1B phosphatase activity, via the activation of cGMP-dependent protein kinases and the mechanism suggested above, reducing CrkII dephosphorylation by PTP1B and thus limiting the downstream effects of ExoT's ADPRT activity.

The antagonistic effect of ExoY-derived cGMP on the cytotoxicity of ExoT is, however, complex and depends on the type of host cells infected and on the predominant mechanism used by ExoT to induce its toxic effects (GAP vs ADPRT). For instance, in A549 cells, where the cytotoxicity of ExoT relies on its GAP activity, we found that the injected ExoY did not produce cGMP. Likely, as with the ADPRT activity of ExoS in NCI-H292 cells, the GAP activity of ExoT in A549 cells may have disrupted the ExoY's activator and consequently cGMP production. As a result, ExoY had no protective effect against ExoT cytotoxicity in these cells. In HeLa cells, as the cytotoxicity of ExoT depends on both its GAP and ADPRT activities, ExoY has only a limited protective effect. The conditions determining whether ExoT cytotoxicity depends primarily on its GAP or ADPRT activity remain to be elucidated. One determinant could be the expression of CrkII in the cells. When CrkII is expressed, ExoT can induce cell retraction by either or both of its activities, as seen in the infection of NCI-H292 and HeLa cells. Conversely, when CrkII is not expressed, ExoT can induce cell rounding solely by its GAP activity, as seen in the infection of A549 cells.

We previously showed that the presence of a T3SS effector gene does not guarantee that it will be injected and activated in the host target cells[30]. Our present study now demonstrates that the type of host cells targeted, the combination of T3SS effectors injected and their mechanism used to induce cytotoxicity can also affect effector activation. Indeed, ExoY is inactive when co-injected with ExoS and also potentially when the cytotoxicity of ExoT depended specifically on its GAP activity. Our findings reveal that ExoS and ExoT not only exert their own cytotoxic effects, but also play a critical role in ExoY activation and, consequently, in their own feedback regulation by ExoY-synthesized cGMP. Our description of how ExoY activity is modulated by bacterial and cellular context (T3SS effectors co-injected and cell type infected) could be a plausible explanation for

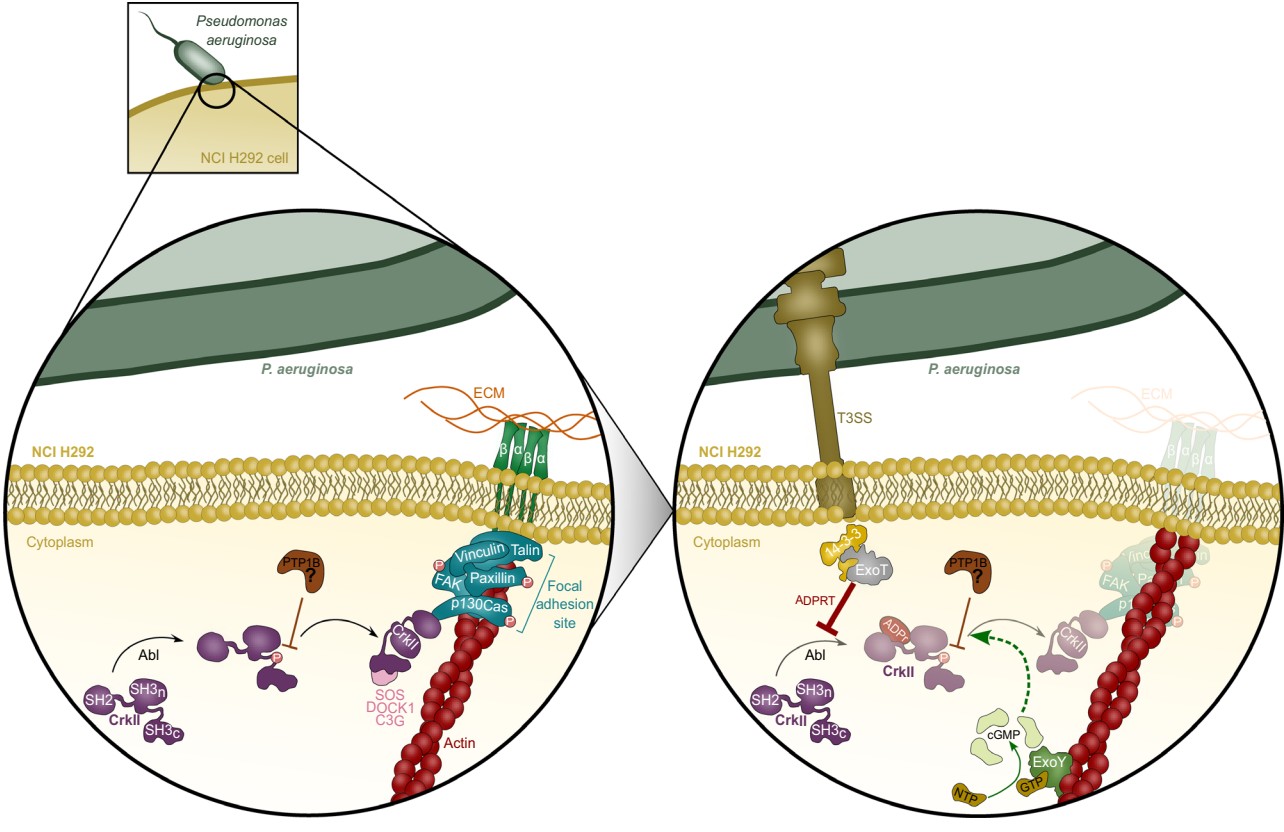

**Fig. 5 | Proposed model of the interplay between ExoY cGMP production and ExoT ADPRT activity in NCI-H292 cells.** Prior to infection, CrkII is phosphorylated by Abl to localize to focal adhesion sites where the adaptor protein is dephosphorylated by a still unknown phosphatase, potentially PTP1B, to interact with tyrosine-phosphorylated p130Cas and other partners (thin brown line). Upon delivery of ExoT and ExoY to the cell cytoplasm by T3SS, ExoY binds to F-actin to become activated and produce cNMPs, primarily cGMP. At the same time, the ADPRT domain of ExoT is activated by the interaction of ExoT with cytoplasmic 14-3-3 proteins. Active ExoT ADP-ribosylates CrkII, which prevents its interaction with Abl (thick red line) and thus the phosphorylation of CrkII by Abl. This in turn prevents the localization of CrkII to the membrane. Whether the prevention of CrkII phosphorylation by ExoT is sufficient to explain the rapid dephosphorylation of the protein is currently unknown. In addition, the ADPRT domain of ExoT could indirectly lead to the dephosphorylation of CrkII by a mechanism involving the activation of PTPB1 or another phosphatase. In either case, dephosphorylation and ADP-ribosylation of CrkII prevent the interaction between CrkII and downstream signaling molecules, thereby disrupting focal adhesion sites, and eventually leading to a programmed apoptotic cell death called anoikis, which involves the loss of interaction with the ECM. However, the production of cGMP by ExoY limits the dephosphorylation of CrkII observed in the presence of ExoT, as suggested by the WB analysis of p-CrkII. This limitation could be due to inhibition of ADPRT activity on the Abl/CrkII interaction and/or on the dephosphorylation by the unknown enzyme (dotted green arrow). Therefore, cGMP production by ExoY attenuates ExoT cytotoxicity and thus regulates *P. aeruginosa* virulence. Abl Abelson's tyrosine-protein kinase, PTP1B protein-tyrosine phosphatase 1B, T3SS Type 3 Secretion System, cNMPs cyclic nucleotide monophosphates, ADPRT ADP-ribosyltransferase domain, p-CrkII phosphorylated CrkII, ECM extracellular matrix.

the previously reported discrepancies in its role. This context-dependent activity, which had not previously been studied, may help to reconcile the conflicting data in the literature and highlights the complex regulatory role of ExoY in the pathogenesis of *P. aeruginosa*.

Studying the interplay between multiple effectors is particularly challenging when they trigger overlapping phenotypes, as is the case with ExoS and ExoT, which both induce cell retraction. Previous studies suggested functional crosstalk among *P. aeruginosa* T3SS effectors[46,47,98,99]. However, here, by analyzing ExoY in combination with either ExoS or ExoT, we reveal a previously unrecognized layer of complexity in *P. aeruginosa*'s pathogenic strategy. Our results suggest that co-delivered effectors may engage in dynamic interplay shaped by their activities and the host cell context, potentially leading to regulatory mechanisms more intricate than previously recognized. Our results were obtained during the study of infection by extracellular bacteria, but it will be interesting to explore this interaction in the context of intracellular bacterial infection[43].

To conclude, this work reveals a critical role for ExoY-synthetized cGMP in modulating T3SS effector cytotoxicity, particularly that of ExoT, and highlights how *P. aeruginosa* might fine-tune its virulence to potentially optimize host colonization or establish and/or maintain chronic host infection. Interestingly, although more than 80% of *P. aeruginosa* strains carry the *exoY* gene[100], the gene is highly variable and may be subject to change in expression or secretion level[30,101–104]. This raises the intriguing possibility that ExoY activity may be either retained to finely tune the activity of other effectors such as ExoT or lost to exacerbate overall bacterial virulence, depending on the environment. *P. aeruginosa* may therefore benefit from a regulatory factor such as ExoY, which may explain the widespread presence of the gene among the strains. These results confirm the importance of studying the ExoY effector under various infectious conditions. Additional unexplored effects of ExoY-produced cNMPs likely exist and remain to be discovered.

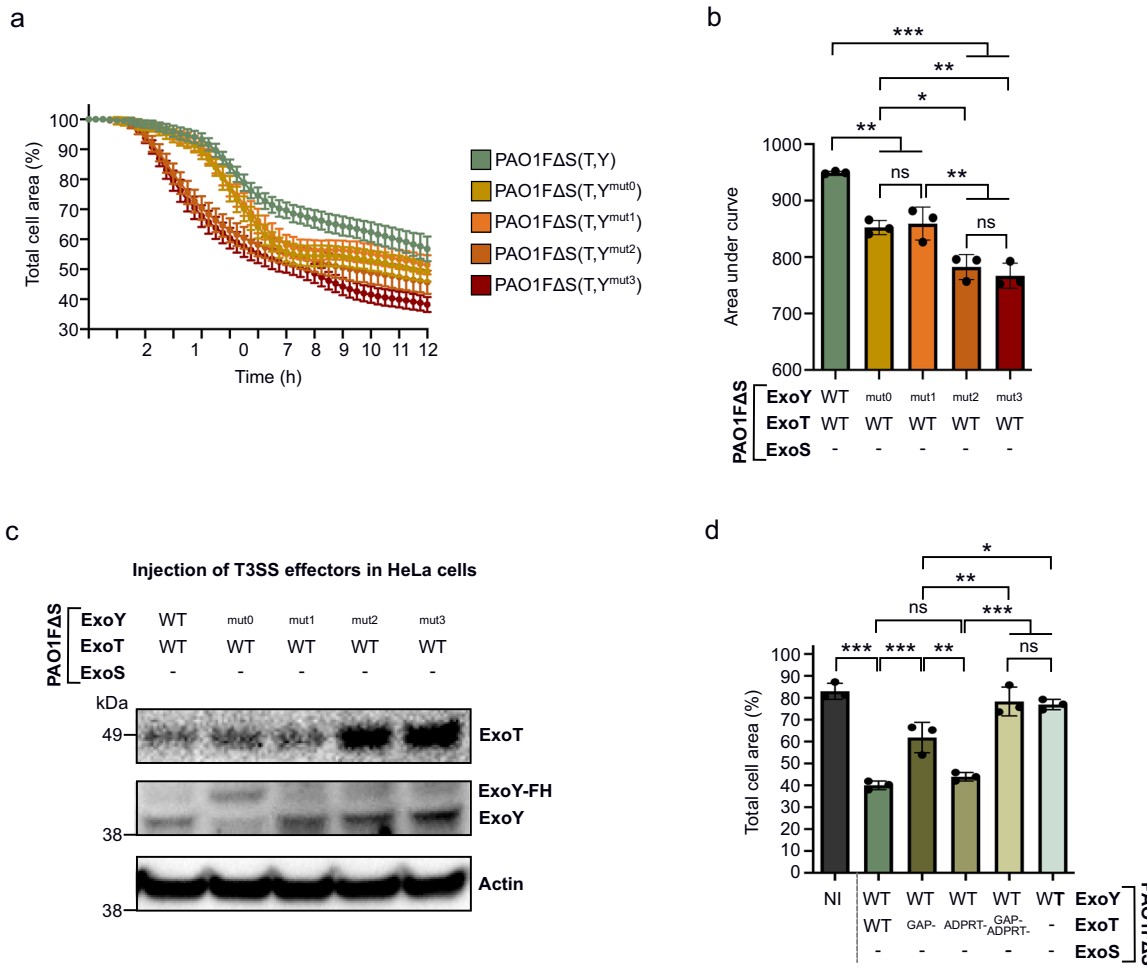

**Fig. 6 | Infection of HeLa cells with PAO1FΔS(T,Y) strains. a** ExoT-induced cell retraction over time in HeLa cells. The results are presented as the mean percentage of the total cell area in comparison to the area before infection. One biological replicate represented as a mean ± SD. **b** Cytotoxicity level of ExoT-expressing strains on HeLa cells. The area under the curve (AUC) for each infection curve was plotted and each dot represents a separate cell retraction assay (biological replicate). **c** Western blot of injected T3SS effectors after infection of HeLa cells with PAO1FΔS(T,Y) strains. Cells were infected for 4 h. The ExoY$^{mut0}$ variant in the PAO1FΔS(T,Y) strain has a higher molecular weight because it contains a Flag-His tag at its C-terminus. **d** Quantification of total cell area of infected HeLa cells. Cells were infected at MOI 50 with PAO1FΔS(T,Y) strains co-injecting WT ExoY with WT ExoT or its inactive mutants. Values indicate the percentage of total cell surface after 10 h of infection as compared to the surface before infection (T 0 h set to 100%). Each data point represents a biological replicate corresponding to the average value of 17 images (technical replicates). '-' means that the effector is not

expressed due to deletion of the corresponding gene. NI means 'not infected'. Data of biological replicates represented as mean ± SD, n = 3 (**b**, **d**). Statistical differences were established by one way ANOVA followed by Tukey's multiple comparison test. Exact P values (**b**): PAO1FΔS(T,Y) vs. PAO1FΔS(T,Y$^{mut0}$) = 0. 001, PAO1FΔS(T,Y) vs. PAO1FΔS(T,Y$^{mut1}$) = 0. 002, PAO1FΔS(T,Y) vs. PAO1FΔS(T,Y$^{mut2}$) or PAO1FΔS(T,Y$^{mut3}$) = <0.001, PAO1FΔS(T,Y$^{mut0}$) vs. PAO1FΔS(T,Y$^{mut2}$) = 0.01, PAO1FΔS(T,Y$^{mut0}$) vs. PAO1FΔS(T,Y$^{mut3}$) = 0.003, PAO1FΔS(T,Y$^{mut1}$) vs. PAO1FΔS(T,Y$^{mut2}$) = 0.006, PAO1FΔS(T,Y$^{mut1}$) vs. PAO1FΔS(T,Y$^{mut3}$) = 0.001, (**d**): NI vs. PAO1FΔS(T,Y) = < 0.001, PAO1FΔS(T,Y) vs. PAO1FΔS(T$^{GAP-}$,Y) = < 0.001, PAO1FΔS(T$^{GAP-}$,Y) vs. PAO1FΔS(T$^{ADPRT-}$,Y) = 0.003, PAO1FΔS(T$^{GAP-}$,Y) vs. PAO1FΔS(T$^{GAP-/ADPRT-}$,Y) = 0.007, PAO1FΔS(T$^{GAP-}$,Y) vs. PAO1FΔST(Y) = 0.01, PAO1FΔS(T$^{ADPRT-}$,Y) vs. PAO1FΔS(T$^{GAP-/ADPRT-}$,Y) or PAO1FΔST(Y) = < 0.001. * p = 0.033, **p = 0.002, ***p < 0.001, n.s., non-significant. Source data are provided as a Source Data file.

## Methods

### Antibodies and reagents

The mouse recombinant monoclonal Crk p38 antibody, targeting CrkI and CrkII was purchased from Abcam (#ab300630, dilution 1:1000). The rabbit Phospho-CrkII (Tyr221) antibody, detecting endogenous levels of CrkII only when phosphorylated at Tyrosine 221, was purchased from Cell Signaling Technology (#3491S, dilution 1:500). The mouse monoclonal antibody against β-actin was from Sigma Aldrich (#A2228, dilution 1:10000). The HRP-labeled detection antibody, IgG Detector Solution V2, was from the TAKARA kit named Western BLoT Rapid Detect v2.0 (# T7122A) and was used at a dilution of 1:3000. Rabbit polyclonal antibodies specific to ExoS and ExoT were obtained from Arne Rietsch and used at a dilution of 1:5000 (Case Western Reserve University School of Medicine). The

specific antiserum for ExoY was obtained from Covalab and created by injecting rabbits with recombinant, truncated ExoY (aa26-223) with a His-tag at its C-terminus. This ExoY antibody was used at a dilution of 1:2000. The Cytotrace Red CMTPX was purchased from Cayman (#20698) and diluted to 1 mM with DMSO. Puromycin (#UP9200-A) and geneticin (#10131027) antibiotics were purchased from EUROMEDEX and Gibco, respectively. The Polybrene was from MERCK MILLIPORE (#TR-1003). The complete EDTA-free protease inhibitor cocktail was from Roche (#11873580001) and the Bradford reagent was from BioRad (Protein Assay Dye Reagent Concentrate, #5000006). Spermine NONOate was purchased from ENZO LIFE SCIENCES (#ALX-430-013-M005) and diluted in water to obtain a 50 mM stock solution. 8-Bromoguanosine 3′,5′-cyclic monophosphate (8-Br-cGMP) was purchased from SIGMA ALDRICH CHIMIE

a

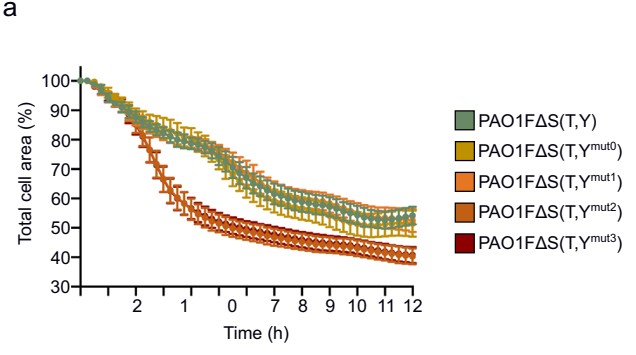

PAO1FΔS(T,Y)
PAO1FΔS(T,Y^mut0)
PAO1FΔS(T,Y^mut1)
PAO1FΔS(T,Y^mut2)
PAO1FΔS(T,Y^mut3)

b

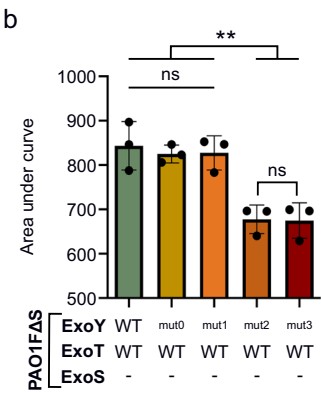

c

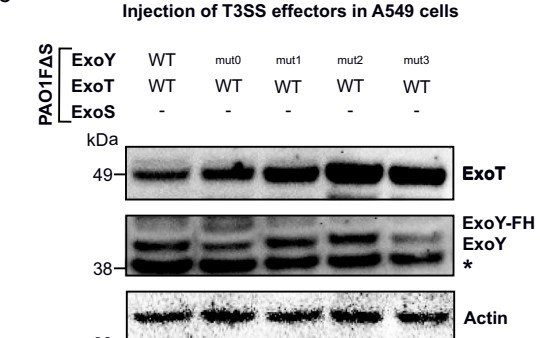

d

| Cells | Average level of cGMP in cells infected with PAO1FΔS(T,Y) strains (pmol/mg of protein) | | | | | |
|---|---|---|---|---|---|---|
| | NI | ExoY^mut0 | ExoY | ExoY^mut1 | ExoY^mut2 | ExoY^mut3 |
| NCI-H292 | 10 ± 8 | 5 ± 2 | 3543 ± 1898 | 5 ± 2 | 3545 ± 1929 | 8 ± 3 |
| A549 | 2 ± 0,1 | 3 ± 0,4 | 93 ± 54 | 2 ± 0,3 | 3 ± 0,5 | 3 ± 1 |

| Cells | Average level of cAMP in cells infected with PAO1FΔS(T,Y) strains (pmol/mg of protein) | | | | | |
|---|---|---|---|---|---|---|
| | NI | ExoY^mut0 | ExoY | ExoY^mut1 | ExoY^mut2 | ExoY^mut3 |
| NCI-H292 | 11 ± 6 | 12 ± 7 | 9 ± 5 | 13 ± 8 | 1686 ± 329 | 94 ± 62 |
| A549 | 10 | 10 ± 2 | 10 ± 2 | 8 ± 2 | 12 ± 3 | 12 ± 0,4 |

e

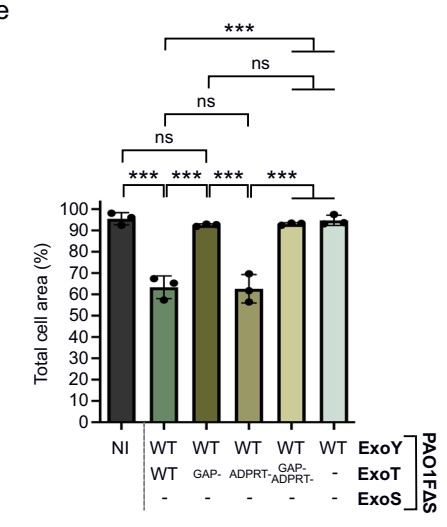

f

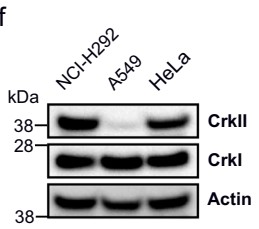

(#B1381-10MG) and diluted in water to obtain a 100 mM stock solution.

## Bacterial strains and growth conditions

The bacterial strains and plasmids used in this study are listed in Supplementary Data 1. Bacterial strains were grown in Luria-Bertani (LB) medium at 37 °C with shaking (180 rpm). Antibiotics were added at the following concentrations: Ampicillin at 100 μg mL−1, gentamycin at 15 or 30 μg mL−1 for *E. coli* and *P. aeruginosa*, respectively. For infection assays, overnight cultures were diluted to optical density (OD$_{600}$) of 0.1 and grown under agitation to reach OD$_{600}$ of 1.

## Eukaryotic cell lines and growth conditions

The human embryonic kidney (HEK) 293 T (ATCC CRL-3216), A549 (ATCC CCL-185), and HeLa (ATCC CCL-2) cells were cultured in Dulbecco's modified Eagle's medium (DMEM) supplemented with 10% fetal bovine serum (FBS). The NCI-H292 cells and their derivatives were cultured in RPMI 1640 medium with GlutaMAX™ supplemented with 10% FBS. Cells transfected with pLVX-IRES-neo vector encoding cAMP or cGMP GloSensor or encoding CrkI or CrkII, were maintained by addition of 200 μg mL−1 Geneticin to the supplemented medium. Cells were grown at 37 °C with 5% CO2 and routinely passaged.

## Plasmid construction

Primers used for PCR amplification, mutations or sequencing are listed in Supplementary Data 2.

pB26 expressing ExoY^K81M-K88I-FH (ExoY^mut0-FH) was created by exchanging the PaeI/Bsp119I fragment from pUM460[14] by that of p1682[22].

**Fig. 7 | Infection of A549 cells with PAO1FΔS(T,Y) strains. a** ExoT-induced cell retraction over time in A549 cells. Cells were infected at MOI of 10 with PAO1FΔS(T,Y) strains. The results are presented as the mean percentage of the total cell area in comparison to the area before infection. One biological replicate represented as a mean ± SD. **b** Cytotoxicity level of ExoT-expressing strains on A549 cells. The area under the curve (AUC) for each infection curve shown in (**a**) was calculated and plotted. Each dot represents a separate cell retraction assay (biological replicate). **c** Western blot (WB) showing injected T3SS effectors in A549 cells. Cells were infected with PAO1FΔS(T,Y) strains at MOI 10 for 4 h. The ExoY$^{mut0}$ variant in the PAO1FΔS(T,Y) strain has a higher molecular weight because it contains a Flag-His tag at its C-terminus. (*) represents a non-specific band corresponding to a cellular protein that reacts with our ExoY antibody. **d** Comparative table of intracellular cGMP and cAMP production after infection of NCI-H292 and A549 cells with PAO1FΔS(T,Y) strains. NCI-H292 and A549 cells were infected for 5 h at MOI 20 and for 4 h at MOI 10, respectively. Intracellular cGMP and cAMP productions were quantified using ELISA kits. Values obtained for NCH-H292 cells have been extracted from Fig. 1h and i. The mean cGMP and cAMP accumulation of each sample is shown, together with the standard deviation. **e** Quantification of the total cell area of infected A549 cell. Cells were infected at MOI 10 with PAO1FΔS(T,Y)

strains co-injecting WT ExoY with WT ExoT or its inactive mutants. Values indicate the percentage of total cell surface after 8 h of infection as compared to the surface before infection (T 0 h set to 100%). Each data point represents a biological replicate corresponding to the average value of 7 images (technical replicates). **f** WB analysis comparing the levels of Crk proteins in NCI-H292 cells, A549 cells and HeLa cells. '-' means that the effector is not expressed due to deletion of the corresponding gene. NI means 'not infected'. Data of biological replicates represented as mean ± SD, n = 3 (**b**, **e**). Statistical differences were established by one way ANOVA followed by Tukey's multiple comparison test. Exact P values (**b**): PAO1FΔS(T,Y) vs. PAO1FΔS(T,Y$^{mut2}$) = 0. 003, PAO1FΔS(T,Y) vs. PAO1FΔS(T,Y$^{mut3}$) = 0. 002, PAO1FΔS(T,Y$^{mut0}$) vs. PAO1FΔS(T,Y$^{mut2}$) = 0. 006, PAO1FΔS(T,Y$^{mut0}$) vs. PAO1FΔS(T,Y$^{mut3}$) = 0. 005, PAO1FΔS(T,Y$^{mut1}$) vs. PAO1FΔS(T,Y$^{mut2}$) = 0. 006, PAO1FΔS(T,Y$^{mut1}$) vs. PAO1FΔS(T,Y$^{mut3}$) = 0. 005, (**d**): NI vs. PAO1FΔS(T,Y) = < 0.001, PAO1FΔS(T,Y) or PAO1FΔS(T$^{GAP-}$,Y) = < 0.001, PAO1FΔS(T,Y) vs. PAO1FΔS(T,Y) or PAO1FΔST(Y) = < 0.001, PAO1FΔS(T$^{GAP-}$,Y) vs. PAO1FΔS(T$^{ADPRT-}$,Y) = < 0.001, PAO1FΔS(T$^{ADPRT-}$,Y) vs. PAO1FΔS(T$^{GAP-/ADPRT-}$,Y) or PAO1FΔST(Y) = < 0.001. * p = 0.033, **p = 0.002, ***p < 0.001, n.s., non-significant. Source data are provided as a Source Data file.

pUM717 expressing ExoY$^{N264V-F289V-S292G-D293A}$-FH (ExoY$^{mut1}$-FH), was created by exchanging the SphI/NdeI fragment from pUM460 by that of pUM700 (see below).

For pUM721, site-directed mutagenesis to introduce the mutation specifying the F83L exchange into *exoY* was performed by PCR amplification with primers VD5-F83L_F and VD5-F83L_R on template pUM460. Nonmutated plasmid (template) was digested by FastDigest DpnI (ThermoFisher, #FD1703) and the reaction mixture was transformed into DH5α. Plasmid DNA of a verified clone was digested with Bsp119I and SphI to subclone the fragment coding for the F83L mutation into an unmodified pUM460 vector digested with Bsp119I and SphI. Subcloning was performed to exclude potential PCR-introduced mutations in the vector sequence.

For pUM727 (expressing ExoY$^{mut2}$-FH), primers VD11-S292G_F and VD11-S292G_R were used to introduce the mutation coding for the S292G exchange into pUM460 as described above. The Bsp119I/NdeI fragment specifying the S292G mutation was then inserted into the pUM721, already containing the F83L mutation.

For pUM733 (expressing ExoY$^{mut3}$-FH), the E258D mutation was added to pUM727 in the same manner, using the oligonucleotide primers VD13-E258D_F and VD13-E258D_R. The Bsp119I/NdeI fragment, containing the S292G and E258D mutations, was then inserted into the pUM721.

The pVR2 expressing AcGFP under the $P_{TRE3G-BI}$ promoter was created by inserting the BamHI/Not1 fragment of pAcGFP-N1, coding for AcGFP, into MCS1 of pTRE3G-BI. Next, the *exoY* gene of pUM445[14] was amplified by PCR with primers UM245 and UM374 and cloned into EcoRI/MluI-digested MCS2 of pVR2, yielding pUM542.

The expression plasmid pUM546 was constructed by replacing the SmaI/RsrII fragment of p1682[22] (containing the K81M and K88I mutations in ExoY, ExoY$^{mut0}$) into pUM542.

The plasmid pUM700 encoding ExoY with the N264V, F289V, S292G and D293A mutations (ExoY$^{mut1}$) was created in two steps. First, site-directed mutagenesis was used on pUM460, as described previously, with primers VD-N264V_F and VD-N264V_R to introduce the N264V mutation. This was followed by another site-directed mutagenesis using primers VD-F289V-S292G-D293A_F and VD-F289V-S292G-D293A_R to introduce the F289V, S292G and D293A mutations. After digestion of the non-mutated plasmid with DpnI, the sequence-verified mutated plasmid was digested with Bsp119I and NcoI, and the extracted fragment was cloned into pUM542, digested with the same restriction enzymes, to replace the WT fragment of the *exoY* gene.

Transfection plasmid pUM731, encoding *exoY*$^{F83L-S292G}$ (*exoY*$^{mut2}$), was created by exchanging the NcoI/ SphI fragment from pUM542 by that of pUM727.

Similarly, pUM734, encoding *exoY*$^{F83L-E258D-S292G}$ (*exoY*$^{mut3}$), was created by exchanging the NcoI/ SphI fragment from pUM542 by that of pUM733.

pUM517 was created by PCR-amplification of pUM445[14] with UM345 and UM246 and cloning of the fragment into NheI/XhoI-digested pAcGFP-N1.

The construction of pUM549, to introduce by allelic exchange the K81M and K88I mutations (ExoY$^{mut0}$) into the *exoY* of PAO1F(S,T,Y) strains as well as the region coding for a C-terminal Flag-His tag, was carried out in several steps. First, an intermediate plasmid named pB93Gm was created, containing the *exoY* sequence, including the genomic sequences located upstream and downstream of the gene. To this end, a PCR reaction amplifying the genomic sequence upstream of *exoY* was performed on PAO1(S,T,Y) genomic DNA (gDNA) using primers #124 and #125 and incorporated into plasmid pEX18Gm as HindIII/PstI fragment. Subsequently, a fragment containing the genomic sequence downstream of *exoY* was PCR-amplified from gDNA of PAO1(S,T,Y) using primers #126 and #127 and introduced into this plasmid as XbaI/EcoRI fragment. A fragment coding for *exoY* and a C-terminal Flag-His tag was then PCR amplified from pUM460 using primers #128 and #129 and inserted as PstI/XbaI fragment into the previous plasmid yielding pB93Gm. Finally, the sequence coding for the K81M and K88I mutations (mut0) was introduced into pB93Gm by replacing the BamHI/SphI fragment with that of p1682[22] to yield pUM549.

pUM707 was created to introduce *exoY* alleles coding for variants ExoY$^{mut1}$, ExoY$^{mut2}$ and ExoY$^{mut3}$ into PAO1F(S,T,Y) by allelic exchange. To construct pUM707, a 2032-bp fragment was PCR amplified from chromosomal DNA of PAO1FΔS(T,Y) using primers PAO1-Up_F and PAO1-Down_R and inserted into pEX18Gm between the EcoRI and HindIII sites.

NcoI/SphI fragments specifying the mutations in ExoY$^{mut1}$, ExoY$^{mut2}$ and ExoY$^{mut3}$ were then isolated from plasmids pUM700, pUM727 and pUM734 to replace the corresponding fragment in pUM707 yielding pUM712, 728 and 736 respectively.

For the construction of the allelic exchange plasmid used to inactivate the Rho-GAP domain of ExoS (R146K mutation), two PCR fragments were amplified from PAO1F(S,T,Y) chromosomal DNA: PCR1 was performed with primers ExoSR146K-Up_F and ExoSR146K-Up_R, and PCR2 was performed with primers ExoSR146K-Down_F and ExoSR146K-Down_R. Next, a splicing by overlap extension PCR (SOE-PCR) was performed to combine the two PCR fragments using equimolar amounts of PCR1 and PCR2 and the external primers (ExoSR146K-Up_F and ExoSR146K-Down_R). The SOE-PCR fragment obtained was digested with

EcoRI and HindIII to replace the EcoRI/HindIII fragment of pUM707, yielding pUM737.

To generate the plasmid for inactivation of the ADPRT domain of ExoS (E379D and E381D mutations), the same protocol was applied but with primers ExoSE379/381D-Up_F and ExoSE379/381D-Up_R for the PCR1, and the primers ExoSE379/381D-Down_F and ExoSE379/381D-Down_R for the PCR2, resulting in pUM738.

The mutation coding for R149K in *exoT* inactivating its Rho-GAP was introduced into the allelic exchange vector using primer pairs ExoTR149K-Up_F/ExoTR149K-Up_R and ExoTR149K-Down_F/ExoTR149K-Down_R for SOE-PCR and replacing the EcoRI/HindIII fragment in pUM707, yielding pUM739.

The mutations coding for E383D and E385D in *exoT* inactivating its ADPRT domain were introduced into the allelic exchange vector using primer pairs ExoTE383/385D-Up_F plus ExoTE383/385D-Up_R and ExoTE383/385D-Down_F plus ExoTE383/385D-Down_R for SOE-PCR and inserted as BamHI/HindIII fragment into pEX18Gm, yielding pUM740.

pUM752 was created to replace the ADPRT domain of ExoT by that of ExoS. To this end, three PCR fragments were amplified from PAO1F(S,T,Y) chromosomal DNA: PCR1 was performed with primers VD17_F and VD17_R (amplification of *exoS* ADPRT domain), PCR2 was performed with primers VD18_F and VD18_R (amplification of *exoT* GAP domain) and PCR3 was performed with primers VD19_F and VD19_R (amplification of the downstream *exoT* sequence). The fusion of the three PCR fragments was performed in two steps. First equimolar amounts of PCR fragments 1 and 2 were used for amplification with the outside primers VD18_F and VD17_R (SOE-PCR1). Then, a second overlapping PCR (SOE-PCR2) was performed, containing SOE-PCR1 and PCR3 at equimolar amounts and primers VD18_F and VD19_R. The obtained fragment was digested with BamHI and HindIII and cloned into pEX18Gm.

## Purification of ExoY toxins

ExoY-FH effectors encoded on pUM460, pB26, pUM717, pUM727 and pUM733 were expressed from the $\lambda P_L$ promoter controlled by the temperature-sensitive cI repressor (cI857) in *E. coli* BLR. 400 mL cultures in LB were started at an OD of 0.05 and incubated for 2h30 at 30 °C before shifting the temperature to 37 °C for 3 h to express the proteins. Proteins were purified from the insoluble protein fraction through their C-terminal His-tag using Ni-NTA chromatography under denaturing conditions (in the presence of 8 M urea). Proteins were renatured by dialysis into 20 mM Tris-HCl pH 9.0, 500 mM NaCl, 10% glycerol, 1 mM 1,4-dithiothreitol (DTT) and 0.4 mM phenylmethylsulfonyl fluoride (PMSF).

## In vitro quantification of cNMPs synthesis by ExoY proteins

Rabbit skeletal muscle α-actin (UniProt P68135) was purified from acetone-dried powder (Pel-Freez Biologicals, Rogers, AR, USA) and ExoY-catalyzed cNMP synthesis was quantified as previously described[14]. Reactions were carried out for 10, 30 or 60 min at 30 °C in the presence of 3 µM of pre-formed F-actin (steady-state polymerized), 5 ng to 1 µg of ExoY proteins, and 2 mM NTP spiked with 0.1 µCi of [α−33P] GTP, 0.1 µCi of [α−33P] ATP, 0.1 µCi of [α−32P] CTP or 0.1 µCi of [α−32P] UTP. Radioactivity (33P or 32P) was measured using a TriCarb scintillation counter (Perkin Elmer). All reactions were performed in triplicates.

## Allelic exchange on P. aeruginosa strains

Mutations on *exoS, exoT*, and *exoY* genes, on chromosomal DNA of PAO1F(S,T,Y) strains, were incorporated using a two-step allelic exchange protocol as described before[105]. Sucrose resistant allelic exchange mutants obtained after the second cross-over event were screened using colony PCRs with Amplif-PAO1 primers for amplification of the fragment and Seq-PAO1 primers for sequencing, as

specified in Supplementary Data 2. It should be noted that the generation of GAP⁻/ADPRT⁻ double mutants on the *exoS* and *exoT* genes were carried out in two steps. First, clones containing the GAP⁻ mutation were selected, then pUM738 or pUM740 allelic exchange vectors were used on these clones to generate the inactivation of the second domain. For the allelic exchange replacing the ADPRT domain of ExoT with that of ExoS, the primers Amplif-PAO1-ExoS-ADPRT_F and Amplif_PAO1-ExoT_R were used to verify the presence of the modified allele. Then PCR amplification from the selected colonies was performed using Amplif-PAO1-ExoT_F and Amplif-PAO1-ExoT_R, followed by sequencing using Seq-ExoT_F and Seq3-ExoS_F.

## In vitro quantification of cNMPs synthesis by ExoY proteins from supernatants

Overnight cultures were diluted into 50 mL of LB with 5 mM EGTA and 20 mM MgCl₂ at OD = 0,05. The cultures were then incubated at 37 °C 180 rpm until they reach an OD of 1,5 after which 1 ml of bacterial culture was sampled for each strain. The samples were centrifuged at 13,000 rpm for 4 min at 4 °C to precipitate the bacteria, and the supernatants were collected. Each supernatant was then mixed with 10 µL of 10% Triton (final concentration 0.1%), rapidly frozen, and stored at −20 °C. Enzymatic activity assays were then performed as previously described[30]. Briefly, for each supernatant, 15 µL was mixed in a 50 µL reaction containing 3 µM Mg-ATP-F-actin polymerized to steady state, 50 mM Tris (pH 8.0), 7.5 mM MgCl₂, 0.5 mg/mL BSA, 200 mM NaCl, 2 mM DTT, and 0.07 mM ATP. After 10 minutes of incubation at 30 °C, reactions were started by adding a radioactive NTP substrate to a final concentration of 2 mM: GTP and ATP were spiked with 0.1 µCi of [α-33P] while UTP and CTP were spiked with 0.1 µCi of [α-32P]. Reactions were incubated for 10, 30 or 120 min at 30 °C and then radioactivity was measured using a TriCarb scintillation counter (Perkin Elmer). The data were finally normalized by the amount of ExoY secreted in the supernatant collected using Western Blot analysis with ExoY standards of known quantities.

## cAMP and cGMP quantification in cells after infection by ELISA

cAMP or cGMP were measured in cells using Direct cAMP/cGMP ELISA kits from Enzo Life Sciences (#ADI-900-066A and #ADI-900-014, respectively). Cells were seeded at $1,5.10^5$ cells per well in a 24-well plate and incubated for 2 days. After infection (see time indicated and MOI used in figure legend), the cell medium was removed from each well and the cells were washed with sterile PBS and frozen. Cells in each well were lysed with 54 µL of 0.1 M HCl solution (supplied with the kit) and incubated for 30 min on ice, then scraped with a pipette tip. The cell lysate was transferred to a 1.5 ml tube and centrifuged for 15 min at 12,000 rpm to remove cell debris. The protein concentration of each supernatant was then analyzed by the Bradford assay and diluted to 0.15 mg/mL. The non-acetylated protocol was followed for preparation of the cAMP and cGMP standards as well as the samples. All samples were measured in duplicate. The yellow color was read at 405 nm.

## Lentiviral particles production

Plasmids pGloSensorTM-22F and pGloSensorTM-40F were digested with NheI and XbaI, and the corresponding genes encoding Promega live-cell biosensors for cAMP and cGMP, respectively, were cloned into the pLVX-IRES-neo vector digested with SpeI and XbaI. Sequencing was performed to select clones with inserts in the desired orientation. Next, lentiviral particles containing the pLVX-neo-cAMP or pLVX-neo-cGMP plasmid were produced. For this, HEK 293 T cells were seeded in two 60 mm Petri dishes, each containing $3.10^6$ cells and 5 mL of cell medium and incubated for 24 h. The cells were then transfected using Lipofectamine 3000 transfection kit (Invitrogen, # L3000008) according to manufacturer's recommendations. Briefly, 14 µL Lipofectamine 3000 and 12.8 µL P3000 Reagent were diluted separately in

two different tubes containing each 500 μL Opti-MEM (Gibco, #31985062). An appropriate mixture of plasmids was added with the diluted P3000 Reagent in a 2:1 ratio of Transfection Reagent: DNA complex. This mixture of plasmids was composed of the pLVX-neo-cAMP or pLVX-neo-cGMP plasmid with the lentiviral packaging and envelope plasmids (psPAX2 and pMD2.G) at a 1:1:1 molar ratio (2.5 μg:2.5 μg:1.4 μg, respectively). The tubes were then mixed and incubated 20 min at room temperature. The transfection mix was added to HEK293T cells in a drop-wise manner and cells were incubated for 6 h. Following this incubation time, the medium of each 60 mm Petri dish was replaced by 5 mL DMEM, containing 1.28% bovine serum albumin (BSA) and 1x Pen/Strep. Transfected cells were incubated again for 72 h, and the lentiviruses-containing media were finally harvested and centrifuged to pellet any packaging cells that were collected during harvesting. Lentiviral particles were stored at −80 °C.

### Establishment of stable cell lines with luminescent GloSensors
NCI-H292 cells were seeded at $3 \times 10^5$ per well in a 6-well plate containing cell medium with 10% FBS and supplemented with 8 μg mL-1 polybrene to promote lentiviral infection. One mL of lentiviral solution, containing the plasmid pLVX-neo-cAMP or pLVX-neo-cGMP was added to the corresponding well. After 24 h incubation, the media was replaced with RPMI containing 10% FBS, supplemented with 800 μg mL-1 Geneticin and cells were regularly passaged until the uninfected cells were dead. The antibiotic concentration was then reduced to 200 μg mL-1. Finally, clones were isolated by limiting dilution and clonal stable populations were selected according to their luminescence emission upon intoxication with CyaA for cAMP-GloSensor cells or transfection (see below with Xfect transfection kit) of plasmids expressing ExoY (pUM517) or ExoY$^{K81M}$ (pUM518) for cGMP-GloSensor cells.

### Luminescence assay
All plasmids for transfection were purified using the NucleoBond Xtra-Midi EF kit (Macherey-Nagel, #740420).

Stable monoclonal cAMP-GloSensor and GMP-GloSensor cells were seeded at $1,5 \times 10^4$ cells per well in a 96-well plate (Corning, #3610), containing RPMI medium supplemented with 200 μg mL-1 Geneticin. Cells were incubated for 2 days before co-transfection with the pEF1α plasmid and the pUM542, pUM546, pUM700, pUM731 or pUM734 plasmids. The co-transfection was performed using the Xfect™ Transfection Reagent (Takara, #631317) and according to manufacturer's recommendations. Briefly, the media of each well was removed and replaced by 100 μL per well of phenol red-free RPMI medium. Next, a mixture was prepared for each condition containing 10 μL of Xfect reaction buffer, 0.15 μg of each plasmid and 0.09 μL of Xfect polymer DNA (0.3 μL / μg DNA). The mixtures were incubated for 10 min at room temperature to allow the nanoparticle complexes to form, and 10 μL of transfection solution was added dropwise to the corresponding well. The plate was incubated at 37 °C 5% $CO_2$ for 6 h before the cell medium was removed and replaced with RPMI supplemented with 200 μg mL-1 Geneticin. The plate was incubated for 2 days at 37 °C 5% $CO_2$ before removing the cell medium and to replace it with RPMI containing 1 μg.mL-1 Doxycycline. After 3 h incubation, the medium in each well was again removed and replaced with 100 μL / well of phenol red-free RPMI containing 5% GloSensor cAMP reagent (Promega, #E1291). Luminescence was monitored every 5 min for 4 h at 24 °C with an automated plate reader (Spark 10 M by TECAN) using an integration time of 1000 ms and the luminescence maximum (see representation on Sup. Fig. 2) was extracted.

For the infection experiments, $2.10^4$ cGMP-GloSensor cells were seeded per well in a 96-well plate for two days. The cell medium was removed, and the cells were infected at an MOI of 20 for 2 h by adding 100 μL of resuspended bacteria in RPMI medium (Gibco, #32404014)

containing GlutaMAX (Gibco, #35050038). Then, 40 μL of resuspended GloSensor cAMP reagent was added to each well to a final concentration of 3.6% and luminescence was recorded every 3 min for 3.5 h at 24 °C.

For CyaA experiments, $2.10^4$ cAMP-GloSensor cells were seeded per well in a 96-well plate for two days. One hour before intoxication, the cell medium was removed and 90 μL of phenol red-free RPMI medium containing GlutaMAX was added per well. This medium was also supplemented with 2 mM $CaCl_2$, 0,5% BSA and 5% GloSensor cAMP reagent. Cells were finally intoxicated with 1 nM final CyaA or CyaA$^{R12E}$ (mutant not translocated, negative control[106]) per well and luminescence was recorded as previously described.

### Knock-out of Crk gene
A pair of oligonucleotides corresponding to a gRNA targeting the *Crk* gene (Supplementary Data 2) was phosphorylated and annealed before being cloned into the Esp3I-digested pLentiCRISPRv2 vector. Then, as previously described, lentiviral particles containing the recombinant plasmid pLentiCRISPR_Crk were created using the HEK293T cells and the psPAX2 and pMD2.G plasmids. NCI-H292 cells were afterwards infected with these lentiviruses using 1 mL of lentiviral particles in a well of a 6-well plate. Transfected cells were selected with 2 μg.mL−1 puromycin for 48 h and clones were isolated by limiting dilution. Clones were selected for their absence of Crk expression by Western blot. One clone was selected for further experiments.

### crk sequence analysis in Crk-deficient cells
Genomic DNA was isolated from $4.10^6$ NCI-H292 and Crk-deficient cells using the DNeasy® Blood & Tissue kit (Qiagen, #69504) and following the manufacturer's protocol. The *crk* gene was amplified by PCR using 100 ng of gDNA from each cell line with primers PCR-Crk_F and PCR-Crk_R (Supplementary Data 2). PCR fragments were then purified and sent for Sanger sequencing with Seq-Crk_F and Seq-Crk_R and TIDE software was used to analyze indels in the *crk* gene from Crk-deficient cells.

### Complementation of Crk-deficient cells
The *CrkI* and *CrkII* genes were synthetized by Twist Bioscience and cloned into the pTwist Amp High Copy plasmid (2221 bp). Some modifications have been introduced, that did not result in changes of the amino acid sequence, to prevent the Cas9 endonuclease from cleaving the gene after its insertion into the genome of deficient cells, and to reduce the %GC content for gene synthesis. Moreover, XbaI and BamHI restriction sites were added at 5' and 3' ends of each gene, respectively (Supplementary Table 1). Each gene was next cloned into XbaI/BamHI-digested pLVX-IRES-neo vector. Lentiviral particles containing the pLVX-neo_CrkI or pLVX-neo_CrkII plasmid were then produced as previously described. Thereafter, Crk-deficient cells were incubated with lentiviral particles under the same conditions as described in the "Establishment of stable cell lines" section, with geneticin selection, and stable cell lines derived from Crk-deficient cells but expressing CrkI or CrkII were selected by limiting dilution. Genomic DNA from each selected clone was extracted using the DNeasy® Blood & Tissue kit (Qiagen, #69504) and primers VD20_F and VD20_R were used for PCR amplification of the modified *crk* gene and sequencing. Selected clones were also tested by WB for CrkI and CrkII expression.

### Cell retraction assay
In our study, cell retraction refers to morphological changes characterized by cytoplasmic reduction and cell rounding in response to bacterial infection. This phenotype was quantified using time-lapse imaging and image analysis of total cell area.

$1.3 \times 10^4$ NCI-H292 cells or their derivatives were seeded per well in a 96-well plate (Greiner, #655090) 48 h before infection ($1.3 \times 10^4$ HeLa

cells or $1.5 \times 10^4$ A549 cells). For the cell retraction assay, cells were first labeled with Cytotrace Red CMTPX (1 μM) for one hour in their FBS-containing culture medium, to label cell cytoplasm after which the medium was removed and replaced with phenol red-free RPMI medium (Gibco, #32404014) supplemented with GlutaMAX (Gibco, #35050038). Due to differences in cell line sensitivity and effector potency, and in order to observe detectable morphological effects during the acquisition period, MOIs varied. Cells were infected at an MOI 20, unless otherwise specified. For example, less cytotoxic strains were used at an MOI of 50, while more sensitive cells (A549 cells) were infected at an MOI of 10. Then, the total cell area was monitored using an IncuCyte live-cell microscope (SX5, Sartorius) containing the Green/Red module. Images from bright field (phase) and red channel (acquisition time 400 ms) were collected every 15 min for 12 h using a x10 objective and treated using the IncuCyte software. Multiple images (2–3) were acquired per well and each condition was tested in multiple wells (typically 8 per experiment). Next, the data from the analysis of images from all wells corresponding to the same condition were averaged to generate a single biological replicate, and the entire experiment was repeated independently to obtain at least three biological replicates per condition.

For the quantification of cell retraction induced by the GAP and ADPRT domains of ExoS and ExoT toxins, the same protocol was used with the exception that the analysis was performed using images acquired with Incucyte (images with the red channel) at a given point and using the Analyze Particles module from FIJI.

### Western Blot (WB) analysis

Cells were seeded at $4.10^5$ cells per well in a 6-well plate and incubated for 2 days ($2.10^4$ A549 or HeLa cells). Next, cells were quantified and infected at the indicated MOI and time, then washed twice with cold PBS before freezing. Cells were lysed with lysis buffer containing 1% Triton X-100, 50 mM Tris-HCl pH 7.4, 150 mM NaCl, 1 mM EDTA, and Roche protease inhibitor cocktail. The lysates were incubated 15 min on ice and centrifuged at $12,000 \times g$ for 15 min at 4 °C and protein concentration in the supernatant was measured using the Bradford assay. Thirty micrograms of protein were then denatured with a buffer containing LDS (Invitrogen, #NP0007) and a reducing agent (Bio-Rad, #1610792) at 95 °C for 5 min. For gel electrophoresis, proteins were run using NuPAGE 4-12% Bis-Tris gels (Invitrogen, #NP0321BOX) and the MES SDS Running buffer (Invitrogen, #NP0002). Proteins were then transferred onto a PVDF membrane, using wet transfer in a transfer cell for 1 h at 4 °C and 350 mA. Finally, the blots were saturated with detergent-containing Tris-buffered saline (TBST) supplemented with 5% BSA. The TBST wash buffer consists of 10 mM Tris pH 7.5, 150 mM NaCl and 0.05% Tween® 20 detergent. The saturation step was carried out for 90 min at room temperature, and the blots were washed three times for 5 minutes each with the TBST buffer before being incubated overnight at 4 °C with the primary antibodies. After three washes, primary antibodies were probed using an HRP-labeled detection antibody for 90 min at room temperature. Then blots were washed three times and signals were detected using the SuperSignal West Pico PLUS Chemiluminescent Substrate (ThermoScientific, #34577) and the ChemiDoc MP Imaging System.

For the visualization of phospho-CrkII at different time points, a similar experiment was performed, but NCI-H292 cells were seeded at $2.5 \times 10^5$ cells per well in a 12-well plate. The cells were then infected at the corresponding times and lysed with a lysis buffer containing 1% Triton X-100, 50 mM Tris-HCl pH 7.4, 150 mM NaCl, 5 mM EDTA, and 1X Halt Protease & Phosphatase Inhibitor Cocktail (Thermo Fisher Scientific, #1861281). When indicated, Spermine NONOate was added to each infected well at 50 μM at the time of infection. Wells infected for 2 h or more received a further 50 μM Spermine NONOate for 2 h. 8-Br-cGMP was added for 5 h at 500 μM to dedicated infected wells (i.e., 1 h and 30 min before the start of infection) and wells with 2 hours and

30 min or more of infection received a further 500 μM 8-Br-cGMP for 2 h and 30 min.

For secretion of T3SS effectors, overnight cultures of the PAO1FΔS(T,Y) and PAO1FΔT(S,Y) strains were diluted to 0.05 and grown for 3 h in LB supplemented with 5 mM EGTA and 20 mM MgCl₂ to reach an OD of 1. Then, 1 ml of each culture was centrifuged to pellet the bacteria and the supernatant was harvested and mixed with 10 μl of 10% Triton X-100 (0.1% final concentration) before being flash frozen. Finally, 14 μL of each of the supernatants (containing T3SS effectors) were denatured with buffer containing LDS and reducing agent and ran using NuPAGE 4-12% Bis-Tris gels for gel electrophoresis. WB analysis of secreted effectors was performed directly from the supernatants, without any concentration step.

### Statistics and reproducibility

GraphPad software (version 9.5.1) was used for statistical analyses. For cytotoxicity assays, AUC was extracted from each retraction curve and plotted on a bar chart. A one-way ANOVA was employed, followed by Tukey's post-hoc test for data comparison. The same statistical method was used to analyze the maximum luminescence of transfected or infected cells expressing cGMP or cAMP GloSensor, the production of cGMP and cAMP in infected cells by ELISA, and the cell retraction induced by ExoS and ExoT domains. For the statistical analysis of guanylate, adenylate, uridylyl and cytidylyl cyclase activities of purified proteins or supernatants, the same method was applied except that a Dunnett's post-hoc test was performed using the sample with WT ExoY as control. Statistical differences in CrkII phosphorylation levels between 2 and 3 h were established using a Fisher's LSD test. Finally, statistical analysis of Fig. 2b was performed using an unpaired two-tailed t-test ($t = 2.84$; $df = 4$) with 95% confidence intervals between 3.15 and 263 and a p-value of 0.05. The number of experimental repetitions for each figure is shown in Supplementary Table 2. The WB shown in Figs. 4d, 4e, 4g, 6c, 7c, and 7f correspond to one representative experiment.

### Reporting summary

Further information on research design is available in the Nature Portfolio Reporting Summary linked to this article.

## Data availability

Source data are provided with this paper.

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

## Acknowledgements

This research was funded, in whole or in part, by the ANR under #ANR-18-CE44-0004 (activExoY, L.R. and U.M.) and under #ANR-23-CE44-0047 (toxi-cUMPsignal, L.R. and U.M.) and by CNRS UMR 3528. A CC-BY public copyright license has been applied by the authors to the present document and will be applied to all subsequent versions up to the Author Accepted Manuscript arising from this submission, in accordance with the grant's open access conditions. V.D. was supported by Pasteur-Roux-Cantarini program and E.L. by a stipend from Amgen Scholars Program. G.D. received a doctoral fellowship from the French Ministry of Higher Education, Research, and Innovation. R.P.L. was supported by a stipend from the Pasteur—Paris University (PPU) International PhD Program. Schematics were created with Inkscape (version 1.2.1). We thank Arne Rietsch (Case Western Reserve University, Ohio), for providing the indicated *P. aeruginosa* strains used in this study and the ExoS and ExoT antibodies. We also thank Éric Faudry and Ina Attrée (IBS, Grenoble) for access to the PAO1F mutant strains. We thank Stefan Dove for his advice on modifying the substrate specificity of ExoY, Alexander Belyy for his preliminary work on ExoY substrate specificity and Martine Comisso for her technical assistance in designing the ExoY variants. We would also like to thank Alexandre Chenal for the CyaA proteins.

## Author contributions

V.D., U.M. and L.R. contributed to the conception and design of the study. V.D. performed all the experiments and analyzed data. G.D. created PAO1F(S,T,Y$^{mut0}$) strain and participated in the ELISA experiments and D.R.B. in the in vitro measurements of cNMPs. E.L. was involved in generating deficient cells and modifying the ADPRT domain of ExoT. V.D. and R.P.L. infected the cGMP reporter cell line. M.T.N. participated in the design of ExoY variants. V.D. wrote the manuscript. V.D., U.M., D.L. and L.R. edited the manuscript. All authors approved the final manuscript.

## Competing interests

The authors declare no competing interests.
