## [Transparent Peer Review file · Nature Communications]

Interplay between T3SS effectors, ExoY activation, and cGMP signaling in *Pseudomonas aeruginosa* infection

Corresponding Author: Mr Vincent Deruelle

Version 1:

Reviewer comments:

Reviewer #1

(Remarks to the Author)

Although present in the vast majority of *Pseudomonas aeruginosa* strains, ExoY is the least understood of the four effectors of the type 3 secretion system (T3SS). In this study, Deruelle and colleagues investigate its role by measuring molecular and phenotypic effects in various cell lines infected by *P. aeruginosa* harboring different ExoY mutants and either of the two effectors ExoS or ExoT. They detect a marked cell type-specific effect of ExoY-mediated changes in cyclic GMP levels on ExoT, but not ExoS activity and feedback effects of ExoS/T on ExoY. The manuscript is exceptionally well-written and the data beautifully presented. Nevertheless, the role of ExoY remains complex and the presented data showing a role in some host cell lines in specific, non-native effector combinations, likely does not fully explain the reason for the conserved presence of ExoY. Furthermore, the manuscript would benefit from additional controls, see below.

Major points:

1. While using the ExoS/T mutant strains to specifically study the effect of ExoY on the remaining effector is a good idea, the authors do not show data showing the effect of ExoY in the wild-type strain harboring both effectors. This data should be shown to allow to evaluate the relevance of the described effects in real-life infections.
2. An important control for the importance of cGMP manipulation and possible independent roles of ExoY would be to influence cGMP levels in the eukaryotic cells independently of ExoY and test if this can override the effects of ExoY / mimic the phenotype in absence of ExoY.
3. The measured effects are relatively minor and restricted to one cell line. The authors should comment on whether this can explain the conserved presence of ExoY? Relatedly, is the described phenotype in line with refs. 21, 32-40 mentioned in the introduction?

Minor points:

1. Can the effects on cGMP levels shown in Fig. 1 also be measured when using the bacteria used in the rest of the manuscript, where the mutants were introduced into the genome?
2. How do the authors explain the hypertranslocation of ExoT into A549 and, even more strikingly, HeLa cells in strains expressing ExoY mut2/3? Is it a true hypersecretion phenotype or a does the stability of ExoT differ? At least in NCI-H292 cells, the ExoY levels seem to differ as well, although this is not correlated with the effect on the eukaryotic cells.
3. Why does ExoYmut2 influence cGMP levels in absence of ExoT, but not in absence of ExoS (Fig. 2G)?

Reviewer #2

(Remarks to the Author)

The conclusion of this study are that ExoY-mediated modulation of cGMP levels impacts ExoT activity, and that conversely ExoT and ExoS can both inhibit the cGMP activity of ExoY. The data supporting this were obtained using both biochemical and cell-based infection assays, utilizing recombinant proteins, and strains of *P. aeruginosa* expressing catalytically active/inactive mutants in Type III toxins. Engineered stable transfected cell lines were used to study intracellular cGMP levels in real time. The cell infection assays were done at the end of the study to assess in-cell relevance, with retraction as the outcome measure for impact on cell intoxication.

Strengths of the study include the reporting of a previously unknown role for ExoY, an understudied effector compared to others encoded by *P. aeruginosa*. Other strengths include the meticulous experimentation using appropriate molecular tools

to engineer bacteria and make toxin variants, and use of transfection systems to engineer host cells to express bacterial proteins and quantify intracellular cGMP levels and their regulation by bacterial toxins. Unfortunately, the authors have overlooked a vast amount of previously published information about *P. aeruginosa*-host cell interactions, which impacts the reliability of the outcome measures used, muddies interpretation of their results, and reduces novelty. For example, it is already known that ExoS can modify its own activity and potentially that of ExoT. Thus, finding that *P. aeruginosa* T3SS effectors can regulate each other's activities is not novel in general. Secondly, the study does not take into account or even mention most of the events occurring when this ExoS producing *P. aeruginosa* strain (PAO1) infects cells, assuming that the only event of relevance is T3SS injection by extracellularly located bacteria. Yet many papers over the past 3 decades have shown that it invades cells, diversifies intracellularly, activates its T3SS inside cells, and secretes T3SS toxins locally inside the cell. The intracellular bacteria utilize the T3SS toxins (primarily ExoS) for trafficking inside the cell, including using ExoS to exit vacuoles, to avoid autophagy, and to construct and then traffic to plasma membrane blebs that can subsequently disconnect allowing a mechanism for extrusion. ExoS also suppresses inflammasome-mediated programmed host cell death, used as a strategy to support survival of the intracellular bacteria. ExoY is also able to induce formation of membrane blebs. All of this is accompanied by complex modifications to the host cell that can cause both cell expansion and cell shrinkage that varies over time. This is beyond the capacity of a simple retraction assay as an outcome measure for intoxication. Indeed cell rounding, similar to the retraction method quantified, was commonly used to assess intoxication before the *P. aeruginosa* research community appreciated the complexities around host cell-*P. aeruginosa* interactions and the advent of other readily available tools to examine cytotoxicity in more detail (e.g. biochemical and imaging based). The authors could have used some of the latter to gain more information about mechanisms for the phenomena noted, especially since the various T3SS effector domains all have different enzymatic activities and lack synchrony in their impacts. Instead, the authors provided a speculative proposed mechanism that is overly complicated. Since the cell retraction method used was also poorly described, it also remains unclear how it accounted for detachment of cells from the culture dish that can occur both dependently or independently of the T3SS e.g. due to proteases. Thus, this study provides only an incremental advance in our understanding of *P. aeruginosa* pathogenesis, despite the results being potentially interesting with aspects done using meticulous experimentation.

Specific comments:

1. The authors state that "each data point in total cell area graphs" are averaged of 16 or 17 images. They also state that 16 or 17 images acquired at 10X magnification from a 96 well plate were averaged for each datapoint. As the imaging field of a 10X objective is large and the total area of a single 96 well plate is small, is it possible that the authors also averaged data from multiple wells in a single experiment? Please clarify how cell rounding parameters were quantified, and the acquisition strategy for the fields of view in the methods section.
2. Related to the previous point that the methods are poorly described, it is not clear whether they account for detachment of cells that commonly occurs when cells are infected with *P. aeruginosa*, sometimes due to intoxication otherwise to bacterial proteases. Indeed, Figure S4 shows differences in total cell numbers between conditions and there appears to be no correlation between cell morphology and the intensity of red staining in the cells. Perhaps certain toxin variants caused cells to detach from the plate causing a reduction in cell numbers.
3. The experimental conditions for infection are different between figures and a rationale for this change has not been stated. This is important as changes to MOI and duration of infection can have significant and unexpected effects on effector expression and impact bacterial cell entry into host cells and host cell responses such as death or in this case cGMP, and Crk levels. e.g. Figure 2- 5h MOI20, Figure 3B- 8h MOI50, Figure 3C- MOI20, Figure 3G- 2h MOI20, Figure 4- 12 hours, Figure 6- 4h, Figure 7- 5h. Methods MOI 20, Figure 3- MOI 50, Figure 4- western blots- 4h MOI50, Figure 6- MOI 50, Figure 7- MOI10, Western blot-4h MOI 10.
4. While the authors initially showed that cGMP levels were modulated by exotoxins in Figure 1, the same experiments were not performed using Crk deficient cells. Thus, it remains unknown if intracellular cGMP levels modulated by the toxins were different in Crk-/- cell lines and if complementation restored this phenotype. While Crk may be a target of ExoT activity, it is not clear how ExoY dependent cGMP regulated this phenotype (either ExoT activity or Crk dephosphorylation) and no direct data is presented in this regard. Is it known whether the rate of Crk dephosphorylation is regulated by cGMP levels? Could this be performed in an in vitro assay? A concurrent quantification of cGMP levels is required to conclude that ExoY-dependent cGMP regulates ExoT function and/or that ExoT directly impacts cell responses.
5. As differences in cell numbers at the time of sampling can have a drastic impact on the relative abundance of proteins and metabolites and resulting conclusions of the study, all data (cGMP levels, western blots) should be normalized to viable cell counts at the time of sampling or show that there are no differences in cell numbers for each condition in the infection assay when images were taken or cells lysed for biochemical analysis.
6. Line 150- Area Under the Curve measurements are used to suggest that ExoT- induced cell rounding is faster- As AUC measures the difference over the entire time period and not at a single time point, it may be more accurate to use the word "greater" while describing differences between groups. Performing statistical analysis of kinetic data will help assert whether ExoT- induced cell rounding is, in fact faster. This should be revised through the manuscript and figures.
7. The controls to test the stated hypothesis are in separate figure panels. e.g. To conclude that only ExoS and not ExoT impacts ExoY activity Figure 3G- Needs to include an ExoS only control with statistical analysis to test this hypothesis, presently only in Figure 3H.

8. A dominant nonspecific band in ExoY western blots is seen only in some conditions, e.g., Figure S3A vs S3B. Could this be due to differences in protein loading between experiments, exposure times for development, or a problem with the ExoY-FH construct that self cleaves if overexpressed? Western blots need to be quantified using densitometry. Additionally, some blots are not clear or missing loading controls Figure S3A (loading control). Figure 7- three bands of ExoY, which of these has been used for quantification?

9. Figure 5 as currently presented implies that only the extracellular bacteria introduce T3SS toxins into the host cell. This ignores that the strain used enter epithelial cells as do most *P. aeruginosa* strains that naturally encode ExoS (i.e. most strains), and that the daughter cells of replicating intracellular bacteria consistently express the T3SS when in the cell cytoplasm. It is important to note here that the studies done by the Barbieri group studying the direct impacts of the T3SS effectors on host cells were done using strain PA103 to inject them, a strain that differs from PAO1 because it does not naturally encode ExoS and has mutations that both reduce its capacity to invade cells. In addition to requiring modification, figure 5 would be better placed at the end of the manuscript or as an author summary if it is to be included.

Minor comments:

10. Define cell retraction and the exact phenotype this refers to.

11. Use alternative terms for phrases like "described above".

12. Figure 1 G,F- Indicate cell line as header to distinguish from cell free data. Figure 1J is out of order. Consider making it a stand-alone table.

13. On the presented graphs, it is not always clear which specific comparisons are significant. Sometimes significance symbols appear to compare bars, other times one bar is compared to groups of bars (e.g.- Figure 3C,3H,4F, 6B,6D,7B and others). The presentation should be consistent with Figure 1D and relevant comparisons clearly indicated.

14. Interchangeable use of toxin and mutant names throughout the manuscript is confusing. Using nomenclature along the lines of Δ ST (only Y) or Δ T (Toxin SY) would help improve readability.

Reviewer #3

(Remarks to the Author)

Reviewer #4

(Remarks to the Author)

This is a well written and highly impactful paper describing the interplay between Type 3 effectors of *P. aeruginosa*. By altering the catalytic activity of ExoY, the authors were able to observe a relative attenuation of ExoT cell-rounding activity. Interestingly, this attenuation was cell-type dependent and correlated to whether the GAP or ADPRT activities were operational in that specific cell type. This is one of the first papers to attribute a function for ExoY-generated cGMP. The authors also were able to show that cGMP somehow interferes with the ADP-ribosylation/dephosphorylation of CrkII. Overall, the results can provide a plausible explanation for disparate observations reported in the literature regarding ExoY's function and the effects of T3SS intoxication in different cell types. There are a few suggestions and/or questions that should be clarified or further explained in the text. The authors may have a little more room to speculate about these results.

Comments for the author's consideration:

1. Lines 55-58. The origin of PA01F, the strain used in this paper is unclear. The authors reference a review article and acknowledge the contribution of Dr. Rietsch. This reviewer was struck by the apparent high production of the T3SS products (Fig. S3), especially from a proteolytic strain of *P. aeruginosa*. In the materials and methods there is no mention of a concentration step when the supernatants are analyzed in Western blots. Could PA01F refer to a hyper-producing strain because of a mutation in ExsD?

2. Changing the nucleotide specificity of ExoY and then showing the interplay between T3SS factors is a tour de force. The data are robust and statistically significant, however, the authors may consider revising Fig.1 to make it easier for readers to have a good perspective on the various activities. What is the linear range of the assay? The authors present a range of .0001 to 1000 pmol nucleotide. The negative control should be the floor of the assay, or this value considered background. Any reading at or below the negative control would be zero.

Looking at the graphs the approximate values seem to be close to:

pmol cG cA cU cC

ExoY 100 10 5 1

mut 0 0 0 0

mut1 0.01 10 0.1 0.05

mut2 10 100 5 2

mut3 1 100 4 2

The actual numbers are more helpful than the +, - or WT designations in Table J.

Another way to express the nucleotide specificity might be a cGMP/cAMP ratio:

ExoY 10

mut1 0.001

mut2 0.1

mut3 0.01

The cellular or infection assays are reasonable as shown. Considering the importance of Fig.1, making the pattern of activities clearer will definitely increase the impact of the manuscript.

3. Line 120-121. The statement suggests that the authors don't know the limit of detection of the assay? Please clarify.

4. Line 128-129. The authors should provide a reference for work showing that ExoY is never injected alone.

5. Lines 132-140. These statements are true, if the mutations introduced into ExoY are secreted normally and have equal intracellular stability as parental ExoY. Fig S3. is an essential control experiment. There is, however, some variation in protein production in the absence and presence of cells. Mut1 is produced the least and in the ExoS+ strain the flag-his tagged version no longer shifts up? Since all proteins are enzymes small variations in protein can impact the overall, biological outcome. Describing the protein levels as "comparable" may not be accurate. Considering the lack of overt cellular lysis as a readout, it may be worth a short discussion paragraph noting this limitation to the system.

6. lines 160-169. ExoS is overtly cytotoxic in this assay system. Note that the area under the curve is reduced for all ExoS expression strains. In Fig. 3, the MOI for ExoS expression strains was reduced, likely to see reduce the pan toxicity response. So many normal cellular processes are likely usurped upon ExoS introduction that it would be difficult to attribute the cellular response to the inability to activate ExoY. Although stating that ExoS ADPRT activity impedes ExoY activity is accurate (lines 210-212) softening the conclusion as to mechanism may be most consistent with the data shown. The authors should keep the 'hypothesis' wording as in the discussion (lines 361-363).

7. Lines 431-433. Have the authors looked at the variation in *exoY* genes? Some postulated that *P. aeruginosa* was slowly mutating ExoY to get rid of the gene or activity. Could other strains of *P. aeruginosa*, perhaps competing in the environment, be using ExoY to control ExoT in a biologically relevant way? Although some reviewers might call the authors on speculating, there could be some thought provoking aspects to discuss.

8. The technical details are well described in a lengthy materials and methods section. Depending on the specific recommendations of the journal, some of these might have to be incorporated into supplementary information to meet space limits.

Version 2:

Reviewer comments:

Reviewer #1

(Remarks to the Author)

The authors have partially addressed the comments and clarified some points in their revised version. In my view, the revision does not yet fully address two previous major points and one minor point, which I listed below.

Previous major point 1

While using the ExoS/T mutant strains to specifically study the effect of ExoY on the remaining effector is a good idea, the authors do not show data showing the effect of ExoY in the wild-type strain harboring both effectors. This data should be shown to allow to evaluate the relevance of the described effects in real-life infections.

Author response:

We fully agree with Reviewer #1. We created a new recombinant wild type strain co-injecting inactive ExoY in order to study ExoY activity when both effectors, ExoS and ExoT, are coinjected. The new experiments are described in lines 219 to 221 and provided in Fig.3i, j, k. Briefly, when all effectors (ExoS, ExoT, ExoY) are injected together, ExoS is the main contributor to bacterial cytotoxicity. In this configuration, cGMP production by ExoY is greatly reduced, supporting our hypothesis that ExoS likely suppresses ExoY activity. ExoY's ability to antagonize ExoT-dependent cytotoxicity therefore remains functional but may be masked in the presence of ExoS.

Reviewer response:

While the data shown in the adapted Fig. 3 is interesting, it is quite difficult to grasp from the figure and the text in the results part. Indicating the strain background in the brackets at the left or right of the labels is a good idea, but it is not clear what version of which protein is expressed from plasmid or from the genome. Especially the "PAO1F T" bracket in Panel k is confusing (perhaps simply wrong?). Similar for the T strain under the "S" bracket in Panel f. The discussion of the additional results should more clearly highlight that the mild effects in the presence of ExoS and ExoT (like in most strains)

Previous major point 2

An important control for the importance of cGMP manipulation and possible independent roles of ExoY would be to influence cGMP levels in the eukaryotic cells independently of ExoY and test if this can override the effects of ExoY / mimic the phenotype in absence of ExoY.

Author response:

We thank the reviewer for this suggestion. Implementing this control is technically challenging in our current experimental system. First, it is difficult to reproduce the spatiotemporal dynamics and cGMP concentrations generated by ExoY using pharmacological agents. Then, maintaining high levels of cGMP throughout infection would likely require multiple additions of the compound. However, our Incucyte live cell imaging system that we used to quantify cell retraction does not have a built-in injection module, which prevents repeated administration of reagents during the test without disrupting the experiment. For these reasons, we are currently unable to perform such a control. However, as shown in Fig. 4g and 4h, we were able to examine the phenotype induced by ExoY by infecting cells with a mutant strain delivering catalytically inactive ExoY and adding SpermineNONOate or 8-Br-cGMP, compounds known to increase intracellular cGMP levels, throughout the experiment. This experiment was possible because we only performed 5 time points and then observed the effect on ExoT-induced Crkl phosphorylation levels.

Reviewer response:

I appreciate the technical challenges. However, I do not see how the experiments shown in Fig. 4gh address the point. If the authors cannot perform the experiments to more directly show the point, the respective statements should be adapted accordingly.

Previous minor point 3

Why does ExoYmut2 influence cGMP levels in absence of ExoT, but not in absence of ExoS (Fig. 2G)?

Author response:

Actually, this is the opposite: ExoYmut2 does increase cGMP levels in the absence of ExoS (PAO1FΔS strain, i.e., expressing ExoT and ExoYmut2), but does not influence cGMP levels in the absence of ExoT (PAO1FΔT strain, i.e., expressing ExoS and ExoYmut2), as the levels are similar to those observed in non-infected conditions. Our hypothesis, as explained in line 169 and described in lines 205 to 226, is that ExoS ADPRT activity interferes with ExoY activation, maybe through alteration of actin polymerization, and thus with its production of cGMP.

Reviewer response:

This is not what the question referred to. It referred to cGMP levels in ExoY and ExoYmut2, which strongly differ in absence of ExoT (124 vs. 4 pmol/mg), but not in the absence of ExoS (3543 vs. 3545 pmol/mg). More precisely phrased: Why does the mut2 mutation in ExoY influence cGMP levels in absence of ExoT, but not in absence of ExoS (Fig. 2G)?

Reviewer #2

(Remarks to the Author)

In their response and the revised manuscript, the authors acknowledge that *P. aeruginosa* T3SS effectors can impact one another's activities, despite the novelty for ExoY. They have also done a good job of addressing the various technical concerns raised by the reviewers.

Two other concerns are less well addressed:

1) Cell shape is a rudimentary and indirect measure for cytotoxicity irrespective of the use of sophisticated image analysis strategies to evaluate outcomes. Cell shape can be impacted for reasons other than toxicity, and when it follows toxicity the details can vary. Options for addressing this include modifying conclusions so they do not overstate what has been shown.

2) The authors continue to downplay/ignore the significant body of literature published over three decades by multiple investigators showing that *Pseudomonas aeruginosa* strain PAO1 and the majority of clinical isolates can adopt a complex intracellular lifestyle in epithelial and other host cells. Altogether, there have been over 100 publications about the intracellular lifestyle of this pathogen (recently reviewed in Resko et al, *J. Bact.*, 2024). This includes studies done in vivo not just in vitro, in many host cell types, and in multiple in vivo models. These have described mechanisms and shown significance, and especially relevant here the major and nuanced role of the type 3 secretion system and its components. This includes a recent paper in this journal showing cooperation between extracellular and intracellular bacteria in non-transformed human epithelial cells and in an in vivo animal model, and that is also dependent on the type 3 secretion system (<https://doi.org/10.1038/s41467-025-62575-3>). The authors justification for ignoring this aspect of the literature appears to be their impression that the non-polarized and transformed cell lines they used, which also did not originate from the tissues usually infected by this pathogen, are somehow of more significance than relevant polarized mucosal epithelial cells of the types infected in vivo which they refer to as "specialized cells". In fact, this could be discussed as a limitation of their study, especially their use of transformed cells at a time when many other options are available. Cell types they used are known to have mutations that alter their behavior, and investigators at the author's own research institute have shown that mutations in one cell type used (HeLa cells) can impact the outcome of host-microbe studies (Tang et al, *Scientific Reports*, 2021). Implying that specific MOI, long time periods, and antibiotic survival assays are required to show intraepithelial *P. aeruginosa* suggests they have not read the relevant literature and it disrespects other investigators research efforts. With regards to the few words added as a response (lines 60-61 and lines 435-437), they are counter to the concern because they infer that there are only extracellular bacteria when *P. aeruginosa* PAO1 infects their cells. PAO1 can invade the cells used (e.g. Sana et al, 2015 *mBio* cited by the authors above, and Kroken et al, *mBio*, 2018), and data to show a lack of intracellular bacteria in their experimental setup has not been included. This Reviewer does not dispute the importance of extracellular *P. aeruginosa* in the pathogenesis of infection. Nor do I ask the authors to study intracellular *P. aeruginosa* – or to perform any additional experiments. But the intracellular capacity of *P. aeruginosa* in many cell types, including the cells used in the author's own study, needs consideration when proposing a model for *P. aeruginosa* pathogenesis. Omitting it perpetuates a long-disproven dogma. Opening to the possibility that this pathogen does more than inject effectors across the plasma membrane might lead the authors to even more opportunities for their line of research.

Reviewer #3

(Remarks to the Author)

Reviewer #4

(Remarks to the Author)

Deruelle, V. et al.

The modified manuscript submitted by Deruelle et al. has many strengths including a thoughtful response to the initial set of reviewers. The concept of effector interplay is not new, but the authors took a novel approach and very thoroughly explored different aspects of the biology of the *P. aeruginosa* effectors and how they interact with each other in cells. They provide important evidence that the interplay not only depends on the effectors but also on host factors, which may differ between different cell types, cell passage or cellular genotype. The information is new, the model system is unique, and the authors provide strong evidence for the various hypotheses tested. Overall, this is an enormous effort, executed with precision and resulting in new insights into a poorly understood family of bacterial enzymes as it intersects with eukaryotic biology.

Key Observations:

1. The authors developed and used ExoY variants that differ in substrate specificity to explore the complexity of cyclic nucleotide accumulation on cellular biology and effector activity. This was a risky approach but certainly resulted in a novel tool.
2. This novel tool was used to:
 - A. Observe that variants of ExoY with diminished cGMP production exacerbated ExoT mediated toxicity but not that of ExoS.
 - B. The authors showed that ExoY-synthesized cGMP specifically attenuates the ADPRT-mediated cytoskeletal rearrangement by ExoT.
 - C. ExoS ADPRT activity but not that of ExoT, decreases ExoY activity.
3. CRISPR technology and pharmacological intervention were used to determine whether the attenuation of ExoT toxicity by ExoY activity was related to CrkII, the cellular target of ExoT ADP-ribosylation. Crk deficient cells were constructed, verified and complemented with CrkII. Using these tools, the authors were able to confirm that CrkII is specifically targeted by ExoT-ADPRT activity and that the mechanism of ExoY-cGMP attenuation involves the rate of CrkII dephosphorylation. cGMP decreases ExoT mediated cytotoxicity by reducing CrkII dephosphorylation. As an added proof, pharmacologically stimulating cGMP production limited CrkII dephosphorylation and attenuated cell rounding mediated by ExoT.
4. Finally, the authors were able to demonstrate that some of the interplay in cytoskeletal dynamics depended on the cell type. HeLa cells behaved much like the NCI-H292 cell line with ExoT-mediated cell rounding being attenuated by ExoY-cGMP production. A549 cells, however, did not show the same phenotype. This mechanism was shown to involve a dependence on the GAP activity of ExoT (as opposed to ADP-ribosylation activity) and expression levels of CrkII.

Vincent Deruelle
Biochemistry of Macromolecular Interactions Unit
Department of Chemistry and Structural Biology
Institut Pasteur
28 rue du Dr. Roux
75015 Paris
vincent.deruelle@pasteur.fr

To Reviewers

Paris, September 15, 2025

Dear Reviewers,

First of all, I would like to thank you for agreeing to evaluate our manuscript and for taking the time to do so thoroughly. We were delighted to read your general comments on this subject. We performed new experiments as suggested and provided answers to all questions. Therefore, there are now a new Fig. 3 that presents experiments on the role of ExoY activity when ExoT and ExoS effectors are co-injected and a new Supplementary Fig. 2 showing the nucleotidyl cyclase activities in bacterial supernatants of *P.aeruginosa* strains secreting the different ExoY variants.

In this letter, we have answered your comments in a point-by-point manner, indicated in green.

Reviewer #1 (Remarks to the Author):

Although present in the vast majority of *Pseudomonas aeruginosa* strains, ExoY is the least understood of the four effectors of the type 3 secretion system (T3SS). In this study, Deruelle and colleagues investigate its role by measuring molecular and phenotypic effects in various cell lines infected by *P. aeruginosa* harboring different ExoY mutants and either of the two effectors ExoS or ExoT. They detect a marked cell type-specific effect of ExoY-mediated changes in cyclic GMP levels on ExoT, but not ExoS activity and feedback effects of ExoS/T on ExoY. The manuscript is exceptionally well-written and the data beautifully presented. Nevertheless, the role of ExoY remains complex and the presented data showing a role in some host cell lines in specific, non-native effector combinations, likely does not fully explain the reason for the conserved presence of ExoY. Furthermore, the manuscript would benefit from additional controls, see below.

Major points:

1. While using the ExoS/T mutant strains to specifically study the effect of ExoY on the remaining effector is a good idea, the authors do not show data showing the effect of ExoY in the wild-type strain harboring both effectors. This data should be shown to allow to evaluate the relevance of the described effects in real-life infections.

We fully agree with Reviewer #1. We created a new recombinant wild type strain co-injecting inactive ExoY in order to study ExoY activity when both effectors, ExoS and ExoT, are co-injected. The new experiments are described in lines 219 to 226 and provided in Fig.3i, j, k. Briefly, when all effectors (ExoS, ExoT, ExoY) are injected together, ExoS is the main contributor to bacterial cytotoxicity. In this configuration, cGMP production by ExoY is greatly reduced, supporting our hypothesis that ExoS likely suppresses ExoY activity. ExoY's ability to antagonize

ExoT-dependent cytotoxicity therefore remains functional but may be masked in the presence of ExoS.

2. An important control for the importance of cGMP manipulation and possible independent roles of ExoY would be to influence cGMP levels in the eukaryotic cells independently of ExoY and test if this can override the effects of ExoY / mimic the phenotype in absence of ExoY.

We thank the reviewer for this suggestion. Implementing this control is technically challenging in our current experimental system. First, it is difficult to reproduce the spatiotemporal dynamics and cGMP concentrations generated by ExoY using pharmacological agents. Then, maintaining high levels of cGMP throughout infection would likely require multiple additions of the compound. However, our Incucyte live cell imaging system that we used to quantify cell retraction does not have a built-in injection module, which prevents repeated administration of reagents during the test without disrupting the experiment. For these reasons, we are currently unable to perform such a control. However, as shown in Fig. 4g and 4h, we were able to examine the phenotype induced by ExoY by infecting cells with a mutant strain delivering catalytically inactive ExoY and adding SpermineNONOate or 8-Br-cGMP, compounds known to increase intracellular cGMP levels, throughout the experiment. This experiment was possible because we only performed 5 time points and then observed the effect on ExoT-induced CrkII dephosphorylation levels.

3. The measured effects are relatively minor and restricted to one cell line. The authors should comment on whether this can explain the conserved presence of ExoY? Relatedly, is the described phenotype in line with refs. 21, 32-40 mentioned in the introduction?

The effect of cGMP production by ExoY on ExoT cytotoxicity is very strong in NCI-H292 cells, but also present in HeLa cells, although potentially weaker given that in these cells, the GAP activity of ExoY also plays an important role in cell retraction in addition to the ADPRT activity (Fig.6). However, other cells, particularly primary cells, and differentiated airway epithelium, should be tested to determine whether the effects of ExoY are limited to a few cells or whether they can be extended.

The reasons for the conserved presence of ExoY and its usefulness for bacterial pathogenicity are discussed in the Discussion section, lines 438–449. This section has been revised following comments from Reviewer #3 (see Reviewer #3, question 7).

In these references (21, 32-40), the authors used different endothelial cells to analyze the ExoY phenotype, as well as deletion mutants or recombinant strains overexpressing ExoY from a high-expression plasmid and therefore injecting non-physiological amounts of ExoY. Importantly, Munder et al. (41) cautioned that the pathology ascribed to ExoY in studies using high-copy number plasmid was not observed with natural *P. aeruginosa* isolates. Here, we used epithelial cells and modified ExoY at the chromosomal level to preserve the native expression of ExoY, so our results cannot be compared to those of these studies. Our discovery shows that, contrary to previous thinking: that ExoY has a minor effect on cytotoxicity, its effect depends on the cellular and bacterial context, thus refining the “minor” role of ExoY to a “specific” role.

Minor points:

1. Can the effects on cXMP levels shown in Fig. 1 also be measured when using the bacteria used in the rest of the manuscript, where the mutants were introduced into the genome?

Our protocol using radioactive activity to measure cNMPs production cannot be used with lysates from infected cells. We therefore developed luminescent reporter cells for cAMP and

cGMP and used ELISA kits to detect cAMP and cGMP after cell infection (similar kits for detection of cUMP and cCMP do not exist). However, we could use our radioactive assay to measure the levels of cNMPs produced by ExoY variants when they are artificially secreted by the T3SS into the bacterial supernatant. These new results are now presented in Supplementary Fig. 2e to 2h and described in lines 120 to 124 of the manuscript. A new paragraph has been added to the Materials and Methods section for the detection of cNMPs using bacterial supernatants (lines 613-628).

2. How do the authors explain the hypertranslocation of ExoT into A549 and, even more strikingly, HeLa cells in strains expressing ExoY mut2/3? Is it a true hypersecretion phenotype or does the stability of ExoT differ? At least in NCI-H292 cells, the ExoY levels seem to differ as well, although this is not correlated with the effect on the eukaryotic cells.

Currently, we do not have an explanation for why we observed more ExoT in cells when it was co-injected with ExoY^{mut2} or ExoY^{mut3}. We agree that it might be due to the ExoT effector's higher stability or translocation, and we have modified the text accordingly (lines 332–334).

3. Why does ExoYmut2 influence cGMP levels in absence of ExoT, but not in absence of ExoS (Fig. 2G)?

Actually, this is the opposite: ExoY^{mut2} does increase cGMP levels in the absence of ExoS (PAO1FΔS strain, i.e., expressing ExoT and ExoY^{mut2}), but does not influence cGMP levels in the absence of ExoT (PAO1FΔT strain, i.e., expressing ExoS and ExoY^{mut2}), as the levels are similar to those observed in non-infected conditions.

Our hypothesis, as explained in line 169 and described in lines 205 to 226, is that ExoS ADPRT activity interferes with ExoY activation, maybe through alteration of actin polymerization, and thus with its production of cGMP.

Reviewer #2 (Remarks to the Author):

The conclusion of this study are that ExoY-mediated modulation of cGMP levels impacts ExoT activity, and that conversely ExoT and ExoS can both inhibit the cGMP activity of ExoY. The data supporting this were obtained using both biochemical and cell-based infection assays, utilizing recombinant proteins, and strains of *P. aeruginosa* expressing catalytically active/inactive mutants in Type III toxins. Engineered stable transfected cell lines were used to study intracellular cGMP levels in real time. The cell infection assays were done at the end of the study to assess in-cell relevance, with retraction as the outcome measure for impact on cell intoxication.

Thank you for highlighting the strengths of our manuscript. However, we would like to clarify that the retraction assays were not performed only at the end of the study, but rather throughout the study to assess the impact of different toxins on host cells in a physiologically relevant context. In fact, they were conducted early on, as presented in Figure 2 and described in the second paragraph of the Results section.

Strengths of the study include the reporting of a previously unknown role for ExoY, an understudied effector compared to others encoded by *P. aeruginosa*. Other strengths include the meticulous experimentation using appropriate molecular tools to engineer bacteria and make toxin variants, and use of transfection systems to engineer host cells to express bacterial proteins and quantify intracellular cGMP levels and their regulation by bacterial toxins.

We thank the reviewer for recognizing the strengths of our study.

Unfortunately, the authors have overlooked a vast amount of previously published information about *P. aeruginosa*-host cell interactions, which impacts the reliability of the outcome measures used, muddies interpretation of their results, and reduces novelty. For example, it is already known that ExoS can modify its own activity and potentially that of ExoT. Thus, finding that *P. aeruginosa* T3SS effectors can regulate each other's activities is not novel in general.

We appreciate the reviewer's comment and agree that previous studies have shown that ExoS can influence its own activity and potentially affect that of ExoT (Riese et al., JBC, 2002; Ichikawa et al., Cell. Microbio., 2005 ; Cisz et al., J. Bacteriol. 2008 ; Armentrout et al., Mol Microbiol., 2021). These studies have suggested that effector interplay may occur and have now been incorporated in our Discussion section lines 430-431. However, to our knowledge, the specific role of ExoY in such effector crosstalk has not been previously described. Our study provides the first evidence that ExoY activity modulates ExoT-induced cytotoxicity through cGMP production and that ExoY function can, in turn, be restricted by ExoS and ExoT in a context-dependent manner. This ExoY-centered regulatory mechanism expands the current understanding of effector interplay and introduces a novel and previously uncharacterized layer of complexity in *P. aeruginosa* pathogenesis.

Secondly, the study does not take into account or even mention most of the events occurring when this ExoS producing *P. aeruginosa* strain (PAO1) infects cells, assuming that the only event of relevance is T3SS injection by extracellularly located bacteria. Yet many papers over the past 3 decades have shown that it invades cells, diversifies intracellularly, activates its T3SS inside cells, and secretes T3SS toxins locally inside the cell. The intracellular bacteria utilize the T3SS toxins (primarily ExoS) for trafficking inside the cell, including using ExoS to exit vacuoles, to avoid autophagy, and to construct and then traffic to plasma membrane blebs that can subsequently disconnect allowing a mechanism for extrusion. ExoS also suppresses inflammasome-mediated programmed host cell death, used as a strategy to support survival of the intracellular bacteria. ExoY is also able to induce formation of membrane blebs. All of

this is accompanied by complex modifications to the host cell that can cause both cell expansion and cell shrinkage that varies over time.

We appreciate the reviewer's detailed comment and agree that *P. aeruginosa* can adopt an intracellular lifestyle under specific conditions. However, this intracellular lifestyle is relatively rare (Sana et al., 2012 JBC; Sana et al., 2015 mBio; Muggeo et al., 2023, PLoS Pathog.), and is primarily observed in specialized cell types (Swart et al., Nat. Microbiol., 2024) or under particular experimental settings, such as antibiotic protection assays, high MOIs, or prolonged infection times. In our study, we used non-polarized epithelial cell lines (NCI-H292, A549, HeLa cells) under conventional 2D cultures and standard infection conditions, without antibiotic treatment, which are commonly used settings to study *P. aeruginosa* as an extracellular pathogen. Consequently, the cell retraction detected in our study results from ExoT or ExoS activity injected by extracellular bacteria, not intracellular bacteria. The intracellular component, although interesting, is not within the scope of this study. We have clarified in our revised manuscript, lines 60-61 and lines 435-437, that our infection models allow *P. aeruginosa* to be studied as an extracellular pathogen. Moreover, although ExoY-induced bleb formation has been observed, it was also reported under conditions not relevant to our model. This is beyond the capacity of a simple retraction assay as an outcome measure for intoxication. Indeed cell rounding, similar to the retraction method quantified, was commonly used to assess intoxication before the *P. aeruginosa* research community appreciated the complexities around host cell-*P. aeruginosa* interactions and the advent of other readily available tools to examine cytotoxicity in more detail (e.g. biochemical and imaging based). The authors could have used some of the latter to gain more information about mechanisms for the phenomena noted, especially since the various T3SS effector domains all have different enzymatic activities and lack synchrony in their impacts. Instead, the authors provided a speculative proposed mechanism that is overly complicated. Since the cell retraction method used was also poorly described, it also remains unclear how it accounted for detachment of cells from the culture dish that can occur both dependently or independently of the T3SS e.g. due to proteases.

We respectfully clarify that our retraction assay is image-based, performed using the IncuCyte live-cell analysis system. Cells were labeled with a cytoplasmic dye, and cell area was quantified over time through automated image analysis across multiple technical replicates. This has been clarified in the main text (lines 143-144). This method allowed us to monitor dynamic morphological changes, including cell retraction, with high temporal resolution. Since it is a reliable and non-invasive method, the IncuCyte system is still actively used to assess morphological changes in real-time (Deruelle et al., 2021, PMID: 33728169; Wlodkowic et al., 2021, PMCID: PMC8275286).

Importantly, no retraction was observed with the PAO1 Δ ST(Y) strain, confirming that the phenotype (cell retraction) is specifically dependent on T3SS-mediated injection of ExoT and/or ExoS. The apparent drop in signal observed after 9 hours post-infection (Fig. 2a) with this strain results from dye dilution and signal loss, not cell rounding or detachment. The decreased intensity of the dye signal is also evident under non-infected conditions after 8h of acquisition (Supplementary Fig. 4). This phenomenon has been clarified in the main text (lines 148-149).

Regarding the Reviewer #2's comment on effector mechanisms: we fully agree that T3SS effectors have distinct enzymatic activities and may act asynchronously. This is precisely why we analyzed ExoY in combination with either ExoS or ExoT, and then inactivated their respective catalytic domains to dissect their individual contributions to the observed

phenotype. Thus, the model we propose is directly supported by the experimental data presented in our study and is consistent with current scientific knowledge, rather than speculative.

Thus, this study provides only an incremental advance in our understanding of *P. aeruginosa* pathogenesis, despite the results being potentially interesting with aspects done using meticulous experimentation.

We respectfully disagree with Reviewer's statement that "this study provides only an incremental advance". As noted before, our study reveals a previously uncharacterized role for ExoY in modulating the activity of co-injected ExoT effector, introducing a novel layer of complexity in *P. aeruginosa* pathogenesis. We believe this finding represents more than an incremental advance, as it expands the current understanding of effector interplay and highlights ExoY as a regulator within the T3SS effector repertoire.

Specific comments:

1. The authors state that "each data point in total cell area graphs" are averaged of 16 or 17 images. They also state that 16 or 17 images acquired at 10X magnification from a 96 well plate were averaged for each datapoint. As the imaging field of a 10X objective is large and the total area of a single 96 well plate is small, is it possible that the authors also averaged data from multiple wells in a single experiment? Please clarify how cell rounding parameters were quantified, and the acquisition strategy for the fields of view in the methods section.

We apologize for the lack of clarity in the Materials and Methods section. Cell retraction was quantified using the Incucyte system with a 10X objective. Infections were performed in 96-well plates, and for each condition, infections were performed in 8 separate wells (technical replicates), and 2–3 images were acquired per well, resulting in 16–24 images per condition per experiment. These images were analyzed and averaged to generate a single data point (biological replicate). This process was repeated in two additional independent experiments, yielding three biological replicates per condition (as shown in Fig. 3b, Fig. 6d and Fig. 7e). We have now clarified this procedure in the revised Methods section (lines 757-761).

2. Related to the previous point that the methods are poorly described, it is not clear whether they account for detachment of cells that commonly occurs when cells are infected with *P. aeruginosa*, sometimes due to intoxication otherwise to bacterial proteases. Indeed, Figure S4 shows differences in total cell numbers between conditions and there appears to be no correlation between cell morphology and the intensity of red staining in the cells. Perhaps certain toxin variants caused cells to detach from the plate causing a reduction in cell numbers. In our experiments, we did not observe significant cell detachment during the infection period. As shown in Supplementary Fig. 4, cells remained attached but exhibited rounding and retraction in response to ExoS or ExoT activity. The Supplementary Fig. 4 also displays zoomed-in images from single wells for visualization, which may give the impression of differing cell densities, but overall cell numbers were comparable between conditions. Cell retraction was quantified by measuring total cell area over time and normalizing to time zero, which controls for any minor variations in cell density. The apparent reduction in red signal is due to dye dilution and signal loss over time, not cell loss, as this phenomenon was also observed under non-infected conditions (Supplementary Fig. 4, images at 8 hours). In addition, for the other conditions, the intensity of red fluorescence may appear reduced in images merged with a bright field, due to bacterial growth affecting the overall contrast.

3. The experimental conditions for infection are different between figures and a rationale for this change has not been stated. This is important as changes to MOI and duration of infection can have significant and unexpected effects on effector expression and impact bacterial cell entry into host cells and host cell responses such as death or in this case cGMP, and Crk levels. e.g. Figure 2- 5h MOI20, Figure 3B- 8h MOI50, Figure 3C- MOI20, Figure 3G- 2h MOI20, Figure 4- 12 hours, Figure 6- 4h, Figure 7- 5h. Methods MOI 20, Figure 3- MOI 50, Figure 4- western blots- 4h MOI50, Figure 6- MOI 50, Figure 7- MOI10, Western blot-4h MOI 10.

We thank the reviewer for this observation. Indeed, different MOIs and infection times were used depending on the cell line, effector activity, and assay sensitivity. For example, less cytotoxic strains (PAO1ΔS strains injecting ExoT WT or its mutants) required a higher MOI (50) to observe measurable morphological effects, while A549 cells were more sensitive and were consistently infected at MOI 10. In contrast, HeLa and NCI-H292 cells were typically infected at MOI 20. For luminescence-based assays, shorter infections (2h) were necessary to preserve cell shape and ensure reliable signal detection. These choices followed a consistent rationale and have now been clarified in the Methods section (lines 750-754).

4. While the authors initially showed that cGMP levels were modulated by exotoxins in Figure 1, the same experiments were not performed using Crk deficient cells. Thus, it remains unknown if intracellular cGMP levels modulated by the toxins were different in Crk^{-/-} cell lines and if complementation restored this phenotype. While Crk may be a target of ExoT activity, it is not clear how ExoY dependent cGMP regulated this phenotype (either ExoT activity or Crk dephosphorylation) and no direct data is presented in this regard. Is it known whether the rate of Crk dephosphorylation is regulated by cGMP levels? Could this be performed in an in vitro assay? A concurrent quantification of cGMP levels is required to conclude that ExoY-dependent cGMP regulates ExoT function and/or that ExoT directly impacts cell responses.

To our knowledge, there is no direct evidence in the literature indicating that cGMP levels regulate CrkII dephosphorylation at Y221. As shown in Fig. 4e, injection of active ExoY alone does not alter CrkII phosphorylation levels over time, meaning that cGMP production by ExoY does not directly regulate the level of CrkII dephosphorylation. In the Discussion section (lines 384-400), we propose a potential indirect mechanism based on current knowledge of cGMP signaling and Crk regulation, although this remains speculative.

5. As differences in cell numbers at the time of sampling can have a drastic impact on the relative abundance of proteins and metabolites and resulting conclusions of the study, all data (cGMP levels, western blots) should be normalized to viable cell counts at the time of sampling or show that there are no differences in cell numbers for each condition in the infection assay when images were taken or cells lysed for biochemical analysis.

The time points for cGMP quantification and Western blot analysis were carefully chosen to avoid cell loss while allowing sufficient time for effector injection. To account for any minor variation in cell number, cGMP levels were normalized to total protein content, and Western blot signals were normalized to actin. As mentioned above (question 2), infection assays were also designed to maintain comparable viable cell numbers across conditions.

6. Line 150- Area Under the Curve measurements are used to suggest that ExoT- induced cell rounding is faster- As AUC measures the difference over the entire time period and not at a single time point, it may be more accurate to use the word “greater” while describing

differences between groups. Performing statistical analysis of kinetic data will help assert whether ExoT- induced cell rounding is, in fact faster. This should be revised through the manuscript and figures.

We thank the reviewer for the suggestion. To clarify, the statement “ExoT-induced cell rounding was faster” refers specifically to the kinetics shown in Fig. 2c, not to the AUC analysis. The AUC was used separately to assess overall cytotoxicity across strains co-injecting ExoT with different ExoY variants. We have revised the manuscript to clarify this distinction and referenced the relevant figures accordingly (line 151 and line 155).

7. The controls to test the stated hypothesis are in separate figure panels. e.g. To conclude that only ExoS and not ExoT impacts ExoY activity Figure 3G- Needs to include an ExoS only control with statistical analysis to test this hypothesis, presently only in Figure 3H.

The hypothesis tested is that ExoS ADPRT activity restricts ExoY-mediated cGMP production in NCI-H292 cells. In both Fig. 3g and 3h, the PAO1FAST(Y) strain serves as the control, as it reflects the maximum cGMP levels produced by ExoY alone. This consistent control allows for direct comparison across panels. Since the role of ExoS is evaluated and statistically analyzed in Fig. 3h, we believe the data sufficiently support the conclusion without the need to include an additional ExoS-only control in Fig. 3g. However, following Reviewer #1's comment (question 1), we performed a new experiment (shown in Fig. 3k) to compare the effects of ExoY variants in the presence of either ExoT or ExoS or both. In the new graph, the cGMP production in NCI-H292 cells infected with the two strains that co-inject ExoY with either ExoT or ExoS can be compared side by side in the same experiment.

8. A dominant nonspecific band in ExoY western blots is seen only in some conditions, e.g., Figure S3A vs S3B. Could this be due to differences in protein loading between experiments, exposure times for development, or a problem with the ExoY-FH construct that self cleaves if overexpressed? Western blots need to be quantified using densitometry. Additionally, some blots are not clear or missing loading controls Figure S3A (loading control). Figure 7- three bands of ExoY, which of these has been used for quantification?

The nonspecific band consistently appears when probing ExoY in cellular lysates and is likely due to background signal from the antibody on cell lysate, not ExoY degradation or self-cleavage as the nonspecific band also appears under non-infected conditions. In some blots, this band was cropped if clearly separated from ExoY, which can vary depending on gel running time. In addition, equal amounts of protein were loaded in each condition, and actin was used for normalization when quantification was performed. In Fig. 7c, the three ExoY bands are annotated on the blot; however, no quantification was performed for this figure. Regarding Supplementary Fig. 3a, as this shows secreted T3SS effectors in supernatants following T3SS induction, no constitutive protein can serve as loading control as only T3SS effectors can be secreted. However, bacterial cultures were normalized by OD prior to supernatant collection (see line 798) to ensure comparable input between samples.

9. Figure 5 as currently presented implies that only the extracellular bacteria introduce T3SS toxins into the host cell. This ignores that the strain used enter epithelial cells as do most *P. aeruginosa* strains that naturally encode ExoS (i.e. most strains), and that the daughter cells of replicating intracellular bacteria consistently express the T3SS when in the cell cytoplasm. It is important to note here that the studies done by the Barbieri group studying the direct impacts of the T3SS effectors on host cells were done using strain PA103 to inject them, a strain that

differs from PAO1 because it does not naturally encode ExoS and has mutations that both reduce its capacity to invade cells. In addition to requiring modification, figure 5 would be better placed at the end of the manuscript or as an author summary if it is to be included.

As noted above, our infection conditions were specifically designed to study extracellular T3SS effector injection. We did not use methods such as the use of specialized cell types or specific experimental parameters such as gentamicin protection to select for intracellular bacteria. In addition, PA103 strain fails to express ExoY (Yahr et al., 1998. PNAS) and unlike previous studies using this strain with plasmid-based overexpression of effectors, our chromosomally engineered PAO1F strains deliver effectors at more physiological levels. Finally, Fig. 5 summarizes the host pathways modulated by co-injection of ExoY and ExoT in NCI-H292 cells, based on our experimental data. Since these mechanisms vary by cell type, we believe its current placement, concluding the NCI-H292 section, is appropriate.

Minor comments:

10. Define cell retraction and the exact phenotype this refers to.

Cell retraction refers to the progressive reduction in host cells in response to bacterial infection, characterized by cytoplasmic reduction and cell rounding. This morphological change reflects cytoskeletal disruption, particularly actin remodeling, and is a well-established readout of cytotoxicity caused by injection of ExoS and ExoT effectors (Bouillot et al., Infect. Immun., 2015; Maresso et al., JBC, 2004; Huber et al., Cell Mol Life Sci., 2014). In our study, cell retraction was quantified using time-lapse imaging and image analysis of total cell area, providing a dynamic measure of effector activity over the time of infection. This definition is now included in the section describing the cell retraction assay, lines 742-744.

11. Use alternative terms for phrases like “described above”.

We have revised the manuscript to improve clarity (line 113, 506-507 and 734-735).

12. Figure 1 G,F- Indicate cell line as header to distinguish from cell free data. Figure 1J is out of order. Consider making it a stand-alone table.

We have revised Fig. 1h and 1i by adding “NCI-H292 cells” to the headers. We agree with the odd position of Fig. 1j but we did it to save space. We will follow the advice of Editor regarding the introduction of a stand-alone table.

13. On the presented graphs, it is not always clear which specific comparisons are significant. Sometimes significance symbols appear to compare bars, other times one bar is compared to groups of bars (e.g.- Figure 3C,3H,4F, 6B,6D,7B and others). The presentation should be consistent with Figure 1D and relevant comparisons clearly indicated.

We thank Reviewer #2 for this comment and have revised the graphs accordingly.

14. Interchangeable use of toxin and mutant names throughout the manuscript is confusing. Using nomenclature along the lines of Δ ST (only Y) or Δ T (Toxin SY) would help improve readability.

We fully agree with Reviewer #2 and have now used the same and hopefully clearer nomenclature: Δ followed by the initial of the deleted toxin, then in brackets the injected toxins including the modified toxin if applicable, for example: PAO1F Δ S(T, Y^{mut0}).

We have therefore revised the manuscript, figures, and their legends accordingly, as well as the tables.

Reviewer #3 (Remarks to the Author):

Reviewer #4 (Remarks to the Author):

This is a well written and highly impactful paper describing the interplay between Type 3 effectors of *P. aeruginosa*. By altering the catalytic activity of ExoY, the authors were able to observe a relative attenuation of ExoT cell-rounding activity. Interestingly, this attenuation was cell-type dependent and correlated to whether the GAP or ADPRT activities were operational in that specific cell type. This is one of the first papers to attribute a function for ExoY-generated cGMP. The authors also were able to show that cGMP somehow interferes with the ADP-ribosylation/dephosphorylation of CrkII. Overall, the results can provide a plausible explanation for disparate observations reported in the literature regarding ExoY's function and the effects of T3SS intoxication in different cell types. There are a few suggestions and/or questions that should be clarified or further explained in the text. The authors may have a little more room to speculate about these results.

Comments for the author's consideration:

1. Lines 55-58. The origin of PA01F, the strain used in this paper is unclear. The authors reference a review article and acknowledge the contribution of Dr. Rietsch. This reviewer was struck by the apparent high production of the T3SS products (Fig. S3), especially from a proteolytic strain of *P. aeruginosa*. In the materials and methods there is no mention of a concentration step when the supernatants are analyzed in Western blots. Could PA01F refer to a hyper-producing strain because of a mutation in ExsD?

In lines 55–58, we mainly referred to previous studies that used recombinant PA103 strains (PA103ΔexoUexoT) carrying a multi-copy plasmid to express ExoY, which results in non-physiological levels of the effector.

To our knowledge PA01F was not sequenced in comparison to PAO1 to see whether there are potential differences. As we understand, it was originally chosen among several PAO1 isolates to be used as isogenic WT for the creation of mutants of T3SS effectors. We chose to work with the PA01F strain because many effector-deletion variants are already available in this background, facilitating mechanistic dissection of each toxin's cytotoxic effect. Concerning WB assays shown in Supplementary Fig. 3, we did not concentrate supernatants (by TCA precipitation), as the antibodies used were sufficiently sensitive to detect secreted effectors directly. We have revised the Materials and Methods section to clarify this point (lines 802-803).

In addition, we performed WB analysis (see below) on secreted effectors, again without concentrating the supernatants, using the PA01F(S,T,Y) strain used in our manuscript and a PAO1(S,T,Y) strain from the international *P. aeruginosa* panel (Anthony De Soyza et al, Microbiologyopen, 2013 ; Silistre et al., Frontiers in Microbiology, 2021). The WB shows that effector secretion is similar between the two strains at two different growth times, ruling out the possibility that our PA01F(S,T,Y) is a hyper-productive strain compared to the common PAO1(S,T,Y).

Figure 1: WB analysis of secreted T3SS effectors from PAO1 and PAO1F strains. The bacteria were cultured from an OD of 0.05 to the corresponding OD values in LB supplemented with 5 mM EGTA and 20 mM MgCl₂. For each condition, the bacterial culture was sampled, and bacteria were removed by centrifugation. The supernatants were then collected and loaded onto a gel before probing the T3SS effectors with the corresponding antibodies.

2. Changing the nucleotide specificity of ExoY and then showing the interplay between T3SS factors is a tour de force. The data are robust and statistically significant, however, the authors may consider revising Fig.1 to make it easier for readers to have a good perspective on the various activities. What is the linear range of the assay? The authors present a range of .0001 to 1000 pmol nucleotide. The negative control should be the floor of the assay, or this value considered background. Any reading at or below the negative control would be zero. Looking at the graphs the approximate values seem to be close to:

pmol cG cA cU cC
 ExoY 100 10 5 1
 mut 0 0 0 0
 mut1 0.01 10 0.1 0.05
 mut2 10 100 5 2
 mut3 1 100 4 2

The actual numbers are more helpful than the +, - or WT designations in Table J.

Another way to express the nucleotide specificity might be a cGMP/cAMP ratio:

ExoY 10
 mut1 0.001
 mut2 0.1
 mut3 0.01

The cellular or infection assays are reasonable as shown. Considering the importance of Fig.1, making the pattern of activities clearer will definitely increase the impact of the manuscript.

We would like to thank Reviewer #4 for the helpful comments and suggestions. We have modified Fig. 1j accordingly to indicate the activity values of the different variants for each cNMP.

Concerning Fig. 1b-1e, the cNMP values observed for the catalytically inactive ExoY mutant (ExoY^{mut0}), while very low, were consistently higher than our negative “blank” control (the assay performed without any ExoY protein). For each condition, we subtracted the value of this blank control, representing the true background, from the raw data. As a result, the values obtained with ExoY^{mut0}, although low, reflect a measurable signal above the background and cannot strictly be considered as zero or baseline on the graphs. We have clarified this point in

the revised figure legend. However, due to their very low levels, we have entered "0" in the revised Fig. 1j for cNMP production by ExoY^{mut0}.

3. Line 120-121. The statement suggests that the authors don't know the limit of detection of the assay? Please clarify.

The limit of detection of the ELISA kit is known, according to the manufacturer's specifications (Enzo, #ADI-900-066A), the sensitivity is 1.18 pmol/mL when using the non-acylated protocol, which we followed. Our statement was not meant to question the assay's detection limit, but rather to reflect that, in our experimental conditions, cAMP production by the three ExoY variants, including the catalytically inactive ExoY^{mut0}, was below the detection threshold. Indeed, while Fig. 1c and 1g show that ExoY WT and ExoY^{mut1} exhibit adenylate cyclase activity (*in vitro*), the intracellular cAMP levels measured by ELISA after infection remained comparable to non-infected and ExoY^{mut0}-infected cells. This suggests that in the context of infection, cAMP production by ExoY WT and ExoY^{mut1} may be low, falling below the limit of detection of the kit. We have clarified this sentence lines 119-120.

4. Line 128-129. The authors should provide a reference for work showing that ExoY is never injected alone.

We thank the Reviewer #4 for this comment. We have modified the sentence accordingly (lines 131-132)

5. Lines 132-140. These statements are true, if the mutations introduced into ExoY are secreted normally and have equal intracellular stability as parental ExoY. Fig S3. is an essential control experiment. There is, however, some variation in protein production in the absence and presence of cells. Mut1 is produced the least and in the ExoS+ strain the flag-his tagged version no longer shifts up? Since all proteins are enzymes small variations in protein can impact the overall, biological outcome. Describing the protein levels as "comparable" may not be accurate. Considering the lack of overt cellular lysis as a readout, it may be worth a short discussion paragraph noting this limitation to the system.

As mentioned in the legend of Supplementary Fig. 3, the Flag-His tag was present only in ExoY^{mut0} in the PAO1ΔS(T,Y) strain (used in an early construct). Other ExoY variants in both PAO1ΔS(T,Y) and PAO1ΔT(S,Y) strains lack this tag.

We agree with Reviewer #4 that there are some variations in protein expression between conditions with and without host cells. We also agree that protein levels between ExoY and ExoT variants are not strictly "comparable" in NCI-H292 cells. However, these differences do not appear to correspond to the phenotypes observed in host cells. We have revised the sentence (lines 155-156) to reflect this limitation more accurately.

6. lines 160-169. ExoS is overtly cytotoxic in this assay system. Note that the area under the curve is reduced for all ExoS expression strains. In Fig. 3, the MOI for ExoS expression strains was reduced, likely to see reduce the pan toxicity response. So many normal cellular processes are likely usurped upon ExoS introduction that it would be difficult to attribute the cellular response to the inability to activate ExoY. Although stating that ExoS ADPRT activity impedes ExoY activity is accurate (lines 210-212) softening the conclusion as to mechanism may be most consistent with the data shown. The authors should keep the 'hypothesis' wording as in the discussion (lines 361-363).

We understand the reviewer's comment and have modified the sentence to retain the wording of the hypothesis (lines 169-171).

7. Lines 431-433. Have the authors looked at the variation in *exoY* genes? Some postulated that *P. aeruginosa* was slowly mutating *ExoY* to get rid of the gene or activity. Could other strains of *P. aeruginosa*, perhaps competing in the environment, be using *ExoY* to control *ExoT* in a biologically relevant way? Although some reviewers might call the authors on speculating, there could be some thought provoking aspects to discuss.

We thank the Reviewer for highlighting this important point. We did not look at the variation in *exoY* genes. However, as the Reviewer mentioned, *exoY* has indeed been identified as one of the most variable genes in *P. aeruginosa* during long-term infection, suggesting selection against its activity in chronic infection contexts (Gabrielaite, mBio 2020; Wee BMC Genomics 2018). Genomic sequence comparisons also reveal substitutions in *exoY* between different strains, as well as frequent frameshift mutations in the *exoY* locus, leading to truncated or elongated proteins with reduced enzymatic activity (Silistre, Frontiers in Microbiology, 2021, Belyy, JBC, 2018). All of this further reinforce the idea that the *ExoY* function is subject to selection in strains, perhaps to compete in a specific environment. When *ExoY* is present, the effector can help bacteria to fine-tune its overall virulence dependent of the cellular context. We now briefly discuss these evolutionary aspects in the manuscript (lines 441-449).

8. The technical details are well described in a lengthy materials and methods section. Depending on the specific recommendations of the journal, some of these might have to be incorporated into supplementary information to meet space limits.

Finally, we would like to thank again all the Reviewers for their comments and suggestions that have helped us to improve the quality of the manuscript. We hope that the revised manuscript will be found acceptable for publication in *Nature Communications*.

Yours sincerely,

Vincent Deruelle

Vincent Deruelle
Biochemistry of Macromolecular Interactions Unit
Department of Chemistry and Structural Biology
Institut Pasteur
28 rue du Dr. Roux
75015 Paris
vincent.deruelle@pasteur.fr

To Reviewers

Paris, October 28, 2025

Dear Reviewers,

Thank you again for your careful review of our manuscript. We have clarified the points that were still unclear.

In this letter, we have answered your comments in a point-by-point manner, indicated in green.

Reviewer #1 (Remarks to the Author):

The authors have partially addressed the comments and clarified some points in their revised version. In my view, the revision does not yet fully address two previous major points and one minor point, which I listed below.

Previous major point 1

While using the ExoS/T mutant strains to specifically study the effect of ExoY on the remaining effector is a good idea, the authors do not show data showing the effect of ExoY in the wild-type strain harboring both effectors. This data should be shown to allow to evaluate the relevance of the described effects in real-life infections.

Author response:

We fully agree with Reviewer #1. We created a new recombinant wild type strain co-injecting inactive ExoY in order to study ExoY activity when both effectors, ExoS and ExoT, are coinjected. The new experiments are described in lines 219 to 226 and provided in Fig.3i, j, k. Briefly, when all effectors (ExoS, ExoT, ExoY) are injected together, ExoS is the main contributor to bacterial cytotoxicity. In this configuration, cGMP production by ExoY is greatly reduced, supporting our hypothesis that ExoS likely suppresses ExoY activity. ExoY's ability to antagonize ExoT-dependent cytotoxicity therefore remains functional but may be masked in the presence of ExoS.

Reviewer response:

While the data shown in the adapted Fig. 3 is interesting, it is quite difficult to grasp from the figure and the text in the results part. Indicating the strain background in the brackets at the left or right of the labels is a good idea, but it is not clear what version of which protein is expressed from plasmid or from the genome. Especially the "PAO1F Δ T" bracket in Panel k is confusing (perhaps simply wrong?). Similar for the Δ T strain under the " Δ S" bracket in Panel f. The discussion of the additional results should more clearly highlight that the mild effects in the presence of ExoS and ExoT (like in most strains)

We thank the reviewer for this comment. The strain backgrounds in Figures 3f and 3k were confusing and have been replaced. To improve clarity, we have also specified in the figure

legends that ‘-’ indicates that the corresponding effector is not expressed due to gene deletion (this clarification has been added to the Supplementary Figure legends as well). Throughout the manuscript, all modifications of T3SS effectors were performed directly in the genome of the corresponding strains, and no expression plasmids were used. Finally, we have added a sentence in the discussion of the additional results to better highlight the moderate effect of ExoY when co-injected with both ExoS and ExoT (lines 224-226).

Previous major point 2

An important control for the importance of cGMP manipulation and possible independent roles of ExoY would be to influence cGMP levels in the eukaryotic cells independently of ExoY and test if this can override the effects of ExoY / mimic the phenotype in absence of ExoY.

Author response:

We thank the reviewer for this suggestion. Implementing this control is technically challenging in our current experimental system. First, it is difficult to reproduce the spatiotemporal dynamics and cGMP concentrations generated by ExoY using pharmacological agents. Then, maintaining high levels of cGMP throughout infection would likely require multiple additions of the compound. However, our Incucyte live cell imaging system that we used to quantify cell retraction does not have a built-in injection module, which prevents repeated administration of reagents during the test without disrupting the experiment. For these reasons, we are currently unable to perform such a control. However, as shown in Fig. 4g and 4h, we were able to examine the phenotype induced by ExoY by infecting cells with a mutant strain delivering catalytically inactive ExoY and adding SpermineNONOate or 8-Br-cGMP, compounds known to increase intracellular cGMP levels, throughout the experiment. This experiment was possible because we only performed 5 time points and then observed the effect on ExoT-induced CrkII dephosphorylation levels.

Reviewer response:

I appreciate the technical challenges. However, I do not see how the experiments shown in Fig. 4gh address the point. If the authors cannot perform the experiments to more directly show the point, the respective statements should be adapted accordingly.

We thank the reviewer for this comment. Modulating intracellular cGMP levels to mimic the catalytic activity of ExoY is technically challenging. In Figures 4g and 4h, we used compounds that increase intracellular cGMP levels; however, we agree that this approach cannot fully reproduce the phenotype observed with WT ExoY. We have revised the paragraph (lines 301–313) to clarify that the results shown in Figures 4g and 4h are consistent with the idea that cGMP production by ExoY delays ExoT-induced cell rounding.

Previous minor point 3

Why does ExoYmut2 influence cGMP levels in absence of ExoT, but not in absence of ExoS (Fig. 2G)?

Author response:

Actually, this is the opposite: ExoYmut2 does increase cGMP levels in the absence of ExoS (PAO1FΔS strain, i.e., expressing ExoT and ExoYmut2), but does not influence cGMP levels in the absence of ExoT (PAO1FΔT strain, i.e., expressing ExoS and ExoYmut2), as the levels are similar to those observed in non-infected conditions. Our hypothesis, as explained in line 169 and described in lines 205 to 226, is that ExoS ADPRT activity interferes with ExoY activation,

maybe through alteration of actin polymerization, and thus with its production of cGMP.

Reviewer response:

This is not what the question referred to. It referred to cGMP levels in ExoY and ExoYmut2, which strongly differ in absence of ExoT (124 vs. 4 pmol/mg), but not in the absence of ExoS (3543 vs. 3545 pmol/mg). More precisely phrased: Why does the mut2 mutation in ExoY influence cGMP levels in absence of ExoT, but not in absence of ExoS (Fig. 2G)?

Sorry, we misunderstood your prior answer.

Actually, we do not know exactly why ExoY in the presence of ExoS (and absence of ExoT, i.e. PAO1FΔT strain) produces residual amounts of cGMP (124 pmol/mg as compared to ~ 3500 pmol/mg with PAO1FΔS) while no cGMP (4 pmol/mg) is produced by ExoY^{mut2} in the same condition. First, due to the large standard deviations in these measurements, the actual levels of cGMP may not be significantly different between the two ExoY variants. Second, the specific GC activity of purified ExoY^{mut2} is slightly lower than that of purified ExoY (Figure 1b). Although no difference in cGMP production is observed between ExoY and ExoY^{mut2} when they are fully active in target cells after being injected with ExoT, a difference may be detectable when ExoY and ExoY^{mut2} are co-injected with ExoS, which keeps them in a state of low activity by preventing their activation.

Reviewer #2 (Remarks to the Author):

In their response and the revised manuscript, the authors acknowledge that *P. aeruginosa* T3SS effectors can impact one another's activities, despite the novelty for ExoY. They have also done a good job of addressing the various technical concerns raised by the reviewers.

Two other concerns are less well addressed:

1) Cell shape is a rudimentary and indirect measure for cytotoxicity irrespective of the use of sophisticated image analysis strategies to evaluate outcomes. Cell shape can be impacted for reasons other than toxicity, and when it follows toxicity the details can vary. Options for addressing this include modifying conclusions so they do not overstate what has been shown. We thank the reviewer for this comment but would like to clarify that our analysis of cell shape is not based on a single static measurement. To decipher the interplay between the effectors, we quantified cell morphology using time-lapse imaging and analysis of multiple images acquired throughout infection, thereby providing kinetic information on cytotoxicity. This dynamic approach goes well beyond a rudimentary single time-point assessment. In addition, the absence of morphological changes in cells following the deletion of the *exoS* and *exoT* genes in the strains confirms that the ExoS and ExoT effectors specifically induce changes in cell shape, corresponding to their cytotoxic effects. Specific mutations in their catalytic domains indeed abolish their cytotoxic effects.

2) The authors continue to downplay/ignore the significant body of literature published over three decades by multiple investigators showing that *Pseudomonas aeruginosa* strain PAO1 and the majority of clinical isolates can adopt a complex intracellular lifestyle in epithelial and other host cells. Altogether, there have been over 100 publications about the intracellular lifestyle of this pathogen (recently reviewed in Resko et al, J. Bact, 2024). This includes studies done in vivo not just in vitro, in many host cell types, and in multiple in vivo models. These have described mechanisms and shown significance, and especially relevant here the major

and nuanced role of the type 3 secretion system and its components. This includes a recent paper in this journal showing cooperation between extracellular and intracellular bacteria in non-transformed human epithelial cells and in an in vivo animal model, and that is also dependent on the type 3 secretion system (<https://doi.org/10.1038/s41467-025-62575-3>). The authors justification for ignoring this aspect of the literature appears to be their impression that the non-polarized and transformed cell lines they used, which also did not originate from the tissues usually infected by this pathogen, are somehow of more significance than relevant polarized mucosal epithelial cells of the types infected in vivo which they refer to as “specialized cells”. In fact, this could be discussed as a limitation of their study, especially their use of transformed cells at a time when many other options are available. Cell types they used are known to have mutations that alter their behavior, and investigators at the author’s own research institute have shown that mutations in one cell type used (HeLa cells) can impact the outcome of host-microbe studies (Tang et al, Scientific Reports, 2021). Implying that specific MOI, long time periods, and antibiotic survival assays are required to show intraepithelial *P. aeruginosa* suggests they have not read the relevant literature and it disrespects other investigators research efforts. With regards to the few words added as a response (lines 60-61 and lines 435-437), they are counter to the concern because they infer that there are only extracellular bacteria when *P. aeruginosa* PAO1 infects their cells. PAO1 can invade the cells used (e.g. Sana et al, 2015 mBio cited by the authors above, and Kroken et al, mBio, 2018), and data to show a lack of intracellular bacteria in their experimental setup has not been included. This Reviewer does not dispute the importance of extracellular *P. aeruginosa* in the pathogenesis of infection. Nor do I ask the authors to study intracellular *P. aeruginosa* – or to perform any additional experiments. But the intracellular capacity of *P. aeruginosa* in many cell types, including the cells used in the author’s own study, needs consideration when proposing a model for *P. aeruginosa* pathogenesis. Omitting it perpetuates a long-disproven dogma. Opening to the possibility that this pathogen does more than inject effectors across the plasma membrane might lead the authors to even more opportunities for their line of research.

We thank the reviewer for the attention given to the literature. We did not intend to ignore, nor do we deny, the existence of an intracellular lifestyle for *P. aeruginosa*. We have previously clarified this point in the revised version of the manuscript.

The *Nature Communications* paper cited by the reviewer indeed provides valuable insights; however, it relies on the use of antibiotics to eliminate extracellular bacteria, thereby selectively enriching for intracellular bacteria, an experimental setup that differs from the one used in our study. Moreover, the cell type used in that study (corneal epithelial cells) differs from the cell lines employed in our manuscript, which are widely used to investigate extracellular bacteria.

Several studies have reported that the proportion of internalized *P. aeruginosa* under conditions similar to ours is very low, typically falling below 2% (Muggeo *et al.*, 2023, *PLoS Pathog.*; Sana *et al.*, 2012, *JBC*; Sana *et al.*, 2015, *mBio*). This supports our focus on the interplay of T3SS effectors delivered by extracellular bacteria. We agree that exploring the intracellular context under conditions that promote bacterial internalization would be of interest, but such investigations fall beyond the scope of the present study and could be addressed in future work. In the previous version of the manuscript, we notified this point in the Discussion section (lines 434–436).

Reviewer #3 (Remarks to the Author):

Reviewer #4 (Remarks to the Author):

Deruelle, V. et al.

The modified manuscript submitted by Deruelle et al. has many strengths including a thoughtful response to the initial set of reviewers. The concept of effector interplay is not new, but the authors took a novel approach and very thoroughly explored different aspects of the biology of the *P. aeruginosa* effectors and how they interact with each other in cells. They provide important evidence that the interplay not only depends on the effectors but also on host factors, which may differ between different cell types, cell passage or cellular genotype. The information is new, the model system is unique, and the authors provide strong evidence for the various hypotheses tested. Overall, this is an enormous effort, executed with precision and resulting in new insights into a poorly understood family of bacterial enzymes as it intersects with eukaryotic biology.

Key Observations:

1. The authors developed and used ExoY variants that differ in substrate specificity to explore the complexity of cyclic nucleotide accumulation on cellular biology and effector activity. This was a risky approach but certainly resulted in a novel tool.
 2. This novel tool was used to:
 - A. Observe that variants of ExoY with diminished cGMP production exacerbated ExoT mediated toxicity but not that of ExoS.
 - B. The authors showed that ExoY-synthesized cGMP specifically attenuates the ADPRT-mediated cytoskeletal rearrangement by ExoT.
 - C. ExoS ADPRT activity but not that of ExoT, decreases ExoY activity.
 3. CRISPR technology and pharmacological intervention were used to determine whether the attenuation of ExoT toxicity by ExoY activity was related to CrkI/II, the cellular target of ExoT ADP-ribosylation. Crk deficient cells were constructed, verified and complemented with CrkII. Using these tools, the authors were able to confirm that CrkII is specifically targeted by ExoT-ADPRT activity and that the mechanism of ExoY-cGMP attenuation involves the rate of CrkII dephosphorylation. cGMP decreases ExoT mediated cytotoxicity by reducing CrkII dephosphorylation. As an added proof, pharmacologically stimulating cGMP production limited CrkII dephosphorylation and attenuated cell rounding mediated by ExoT.
 4. Finally, the authors were able to demonstrate that some of the interplay in cytoskeletal dynamics depended on the cell type. HeLa cells behaved much like the NCI-H292 cell line with ExoT-mediated cell rounding being attenuated by ExoY-cGMP production. A549 cells, however, did not show the same phenotype. This mechanism was shown to involve a dependence on the GAP activity of ExoT (as opposed to ADP-ribosylation activity) and expression levels of CrkII.
- We sincerely thank the reviewer for the accurate summary of our work, and for the kind remarks regarding the novelty of our findings and the experimental approaches used. We greatly appreciate the reviewer's recognition of the effort in exploring the interplay among *P. aeruginosa* T3SS effectors and their host cell targets.